# Testing Probabilistic Circuits *†

**Yash Pote** Ⓡ **Kuldeep S. Meel**
School of Computing, National University of Singapore

## Abstract

Probabilistic circuits (PCs) are a powerful modeling framework for representing tractable probability distributions over combinatorial spaces. In machine learning and probabilistic programming, one is often interested in understanding whether the distributions learned using PCs are close to the desired distribution. Thus, given two probabilistic circuits, a fundamental problem of interest is to determine whether their distributions are close to each other.

The primary contribution of this paper is a closeness test for PCs with respect to the total variation distance metric. Our algorithm utilizes two common PC queries, counting and sampling. In particular, we provide a poly-time probabilistic algorithm to check the closeness of two PCs, when the PCs support tractable approximate counting and sampling. We demonstrate the practical efficiency of our algorithmic framework via a detailed experimental evaluation of a prototype implementation against a set of 475 PC benchmarks. We find that our test correctly decides the closeness of all 475 PCs within 3600 seconds.

## 1 Introduction

Probabilistic modeling is at the heart of modern computer science, with applications ranging from image recognition and image generation [29, 30] to weather forecasting [3]. Probabilistic models have a multitude of representations, such as probabilistic circuits (PCs) [9], graphical models [19], generative networks [16], and determinantal point processes [20]. Of particular interest to us are PCs, which are known to support guaranteed inference and thus have applications in safety-critical fields such as healthcare [2, 25]. In this work, we will focus on PCs that are fragments of the Negation Normal Form (NNF), specifically DNNFs, d-DNNFs, SDNNFs, and PIs [13]. We refer to the survey by Choi et al. [9] for more details regarding PCs.

Given two distributions $P$ and $Q$, a fundamental problem is to determine whether they are close. Closeness between distributions is frequently quantified using the total variation (TV) distance, $d_{TV}(P, Q) = (1/2)\|P - Q\|_1$, where $\| \cdot \|$ is the $\ell_1$ norm [21, 6]. Thus, stated formally, closeness testing is the problem of deciding whether $d_{TV}(P, Q) \leq \varepsilon$ or $d_{TV}(P, Q) \geq \eta$ for $0 \leq \varepsilon < \eta \leq 1$. Determining the closeness of models has applications in AI planning [13], bioinformatics [31, 33, 35] and probabilistic program verification [15, 23].

Equivalence testing is a special case of closeness testing, where one tests if $d_{TV}(P, Q) = 0$. Darwiche and Huang [13] initiated the study of equivalence testing of PCs by designing an equivalence test for d-DNNFs. An equivalence test is, however, of little use in contexts where the PCs under test

---

*The accompanying tool, available open source, can be found at https://github.com/meelgroup/teq. The Appendix is available in the accompanying supplementary material.

†The authors decided to forgo the old convention of alphabetical ordering of authors in favor of a randomized ordering, denoted by Ⓡ. The publicly verifiable record of the randomization is available at `https://www.aeaweb.org/journals/policies/random-author-order/search` with confirmation code: icoxrj0sNB2L. For citation of the work, authors request that the citation guidelines by AEA for random author ordering be followed.

encode non-identical distributions that are nonetheless close enough for practical purposes. Such situations may arise due to the use of approximate PC compilation [10] and sampling-based learning of PCs [26, 27]. As a concrete example, consider PCs that are learned via approximate methods such as stochastic gradient descent [27]. In such a case, PCs are likely to converge to close but non-identical distributions. Given two such PCs, we would like to know whether they have converged to distributions close to each other. Thus, we raise the question: *Does there exist an efficient algorithm to test the closeness of two PC distributions?*

In this work, we design the first closeness test for PCs with respect to TV distance, called Teq. Assuming the tested PCs allow poly-time approximate weighted model counting and sampling, Teq runs in polynomial time. Formally, given two PC distributions $P$ and $Q$, and three parameters $(\varepsilon, \eta, \delta)$, for closeness($\varepsilon$), farness($\eta$), and tolerance($\delta$), Teq returns Accept if $d_{TV}(P, Q) \leq \varepsilon$ and Reject if $d_{TV}(P, Q) \geq \eta$ with probability at least $1 - \delta$. Teq makes atmost $O((\eta - \varepsilon)^{-2} \log(\delta^{-1}))$ calls to the sampler and exactly 2 calls to the counter.

Teq builds on a general distance estimation technique of Canonne and Rubinfeld [4] that estimates the distance between two distributions with a small number of samples. In the context of PCs, the algorithm requires access to an exact sampler and an exact counter. Since not all PCs support exact sampling and counting, we modify the technique presented in [4] to allow for approximate samples and counts. Furthermore, we implement and test Teq on a dataset of publicly available PCs arising from applications in circuit testing. Our results show that closeness testing can be accurate and scalable in practice.

For some NNF fragments, such as DNNF, no sampling algorithm is known, and for fragments such as PI, sampling is known to be NP-hard [32]. Since Teq requires access to approximate weighted counters and samplers to achieve tractability, the question of determining the closeness of the PCs mentioned above remains unanswered. Thus, we investigate further and characterize the complexity of closeness testing for a broad range of PCs. Our characterization reveals that PCs from the fragments d-DNNFs and SDNNFs can be tested for closeness in poly-time via Teq, owing to the algorithms of Darwiche [11] and Arenas et al. [1]. We show that the SDNNF approximate counting algorithm of Arenas et al. [1] can be extended to log-linear SDNNFs using chain formulas [8]. Then, using previously known results, we also find that there are no poly-time equivalence tests for PCs from PI and DNNF, conditional on widely believed complexity-theoretic conjectures. Our characterization also reveals some open questions regarding the complexity of closeness and equivalence testing of PCs.

The rest of the paper is organized in the following way. We define the notation and discuss related work in Section 2. We then present the main contribution of the paper, the closeness test Teq, and the associated proof of correctness in Section 3. We present our experimental findings in Section 4, and then discuss the complexity landscape of closeness testing in Section 5. We conclude the paper and discuss some open problems in Section 6. Due to space constraints, we defer some proofs to the supplementary Section A.

## 2 Background

Let $\varphi : \{0, 1\}^n \to \{0, 1\}$ be a circuit over $n$ Boolean variables. An assignment $\sigma \in \{0, 1\}^n$ to the variables of $\varphi$ is a *satisfying assignment* if $\varphi(\sigma) = 1$. The set of all satisfying assignments of $\varphi$ is $R_\varphi$. If $|R_\varphi| > 0$, then $\varphi$ is said to be *satisfiable* and if $|R_\varphi| = 2^n$, then $\varphi$ is said to be *valid*. We use $|\varphi|$ to denote the size of circuit $\varphi$, where the size is the total number of vertices and edges in the circuit DAG.

The polynomial hierarchy (PH) contains the classes $\Sigma_1^P$ (NP) and $\Pi_1^P$ (co-NP) along with generalizations of the form $\Sigma_i^P$ and $\Pi_i^P$ where $\Pi_{i+1}^P = \text{co-NP}^{\Pi_i^P}$ and $\Sigma_{i+1}^P = \text{NP}^{\Sigma_i^P}$ [34]. The classes $\Sigma_i^P$ and $\Pi_i^P$ are said to be at level $i$. If it is shown that two classes on the same or consecutive levels are equal, the hierarchy collapses to that level. Such a collapse is considered unlikely, and hence is used as the basic assumption for showing hardness results, including the ones we present in the paper.

## 2.1 Probability distributions

A weight function $\mathtt{w} : \{0, 1\}^n \to \mathbb{Q}^+$ assigns a positive rational weight to each assignment $\sigma$. We extend the definition of $\mathtt{w}$ to also allow circuits as input: $\mathtt{w}(\varphi) = \sum_{\sigma \in R_\varphi} \mathtt{w}(\sigma)$. For weight function $\mathtt{w}$ and circuit $\varphi$, $\mathtt{w}(\varphi)$ is the weighted model count (WMC) of $\varphi$ w.r.t. $\mathtt{w}$.

In this paper, we focus on log-linear weight functions as they capture a wide class of distributions, including those arising from graphical models, conditional random fields, and skip-gram models [24]. Log-linear models are represented as literal-weighted functions, defined as:

**Definition 1.** *For a set $X$ of $n$ variables, a weight function $\mathtt{w}$ is called literal-weighted if there is a poly-time computable map $\mathtt{w} : X \to \mathbb{Q} \cap (0, 1)$ such that for any assignment $\sigma \in \{0, 1\}^n$ :*

$$\mathtt{w}(\sigma) = \prod_{x \in \sigma} \begin{cases} \mathtt{w}(x) & if \quad x = 1 \\ 1 - \mathtt{w}(x) & if \quad x = 0 \end{cases}$$

For all circuits $\varphi$, and log-linear weight functions $\mathtt{w}$, $\mathtt{w}(\varphi)$ can be represented in size polynomial in the input.

**Probabilistic circuits:** A probabilistic circuit is a satisfiable circuit $\varphi$ along with a weight function $\mathtt{w}$. $\varphi$ and $\mathtt{w}$ together define a discrete probability distribution on the set $\{0, 1\}^n$ that is supported over $R_\varphi$. We denote the p.m.f. of this distribution as: $P(\varphi, \mathtt{w})(\sigma) = \begin{cases} 0 & \varphi(\sigma) = 0 \\ \mathtt{w}(\sigma)/\mathtt{w}(\varphi) & \varphi(\sigma) = 1 \end{cases}$

In this paper, we study circuits that are fragments of the Negation Normal Form (NNF). A circuit $\varphi$ in NNF is a rooted, directed acyclic graph (DAG), where each leaf node is labeled with true, false, $v$ or $\neg v$; and each internal node is labeled with a $\wedge$ or $\vee$ and can have arbitrarily many children. We focus on four fragments of NNF, namely, Decomposable NNF(DNNF), deterministic-DNNF(d-DNNF), Structured DNNF(SDNNF), and Prime Implicates(PI). For further information regarding circuits in NNF, refer to the survey [14] and the paper [28].

The TV distance of two probability distributions $P(\varphi_1, \mathtt{w}_1)$ and $P(\varphi_2, \mathtt{w}_2)$ over $\{0, 1\}^n$ is defined as: $d_{TV}(P(\varphi_1, \mathtt{w}_1), P(\varphi_2, \mathtt{w}_2)) = \frac{1}{2} \sum_{\sigma \in \{0,1\}^n} |P(\varphi_1, \mathtt{w}_1)(\sigma) - P(\varphi_2, \mathtt{w}_2)(\sigma)|$.

$P(\varphi_1, \mathtt{w}_1)$ and $P(\varphi_2, \mathtt{w}_2)$ are said to be (1) equivalent if $d_{TV}(P(\varphi_1, \mathtt{w}_1), P(\varphi_2, \mathtt{w}_2)) = 0$, (2) $\varepsilon$-close if $d_{TV}(P(\varphi_1, \mathtt{w}_1), P(\varphi_2, \mathtt{w}_2)) \leq \varepsilon$, and (3) $\eta$-far if $d_{TV}(P(\varphi_1, \mathtt{w}_1), P(\varphi_2, \mathtt{w}_2)) \geq \eta$.

Our closeness testing algorithm Teq, assumes access to an approximate weighted counter $\mathsf{Awct}(\alpha, \beta, \varphi, \mathtt{w})$, and an approximate weighted sampler $\mathsf{Samp}(\alpha, \beta, \varphi, \mathtt{w})$. We define their behavior as follows:

**Definition 2.** $\mathsf{Awct}(\alpha, \beta, \varphi, \mathtt{w})$ *takes a circuit $\varphi$, a weight function $\mathtt{w}$, a tolerance parameter $\alpha > 0$ and a confidence parameter $\beta > 0$ as input and returns the approximate weighted model count of $\varphi$ w.r.t. $\mathtt{w}$ such that*

$$\Pr\left[\frac{\mathtt{w}(\varphi)}{1 + \alpha} \leq \mathsf{Awct}(\alpha, \beta, \varphi, \mathtt{w}) \leq (1 + \alpha)\mathtt{w}(\varphi)\right] \geq 1 - \beta$$

*Tractable approximate counting algorithms for PCs are known as Fully Polynomial Randomised Approximation Schemes (FPRAS). The running time of an FPRAS is given by $T(\alpha, \beta, \varphi) = poly(\alpha^{-1}, \log(\beta^{-1}), |\varphi|)$.*

**Definition 3.** $\mathsf{Samp}(\alpha, \beta, \varphi, \mathtt{w})$ *takes a circuit $\varphi$, a weight function $\mathtt{w}$, a tolerance parameter $\alpha > 0$ and a confidence parameter $\beta > 0$ as input and returns either (1) a satisfying assignment $\sigma$ sampled approximately w.r.t. weight function $\mathtt{w}$ with probability $\geq 1 - \beta$ or (2) a symbol $\perp$ indicating failure with probability $< \beta$. In other words, whenever $\mathsf{Samp}$ samples $\sigma$:*

$$\frac{P(\varphi, \mathtt{w})(\sigma)}{1 + \alpha} \leq \Pr[\mathsf{Samp}(\alpha, \beta, \varphi, \mathtt{w}) = \sigma] \leq (1 + \alpha)P(\varphi, \mathtt{w})(\sigma)$$

*Tractable approximate sampling algorithms for PCs are known as Fully Polynomial Almost Uniform Samplers (FPAUS). The running time of an FPAUS for a single sample is given by $T(\alpha, \beta, \varphi) = poly(\alpha^{-1}, \log(\beta^{-1}), |\varphi|)$.*

In the rest of the paper $[m]$ denotes the set $\{1, 2, \dots m\}$, $\mathbb{1}(e)$ represents the indicator variable for event $e$, and $\mathbb{E}(v)$ represents the expectation of random variable $v$.

## 2.2 Related work

**Closeness testing:** Viewing circuit equivalence testing through the lens of distribution testing, we see that the d-DNNF equivalence test of Darwiche and Huang [13] can be interpreted as an equivalence test for uniform distribution on the satisfying assignments of d-DNNFs. This relationship between circuit equivalence testing and closeness testing lets us rule out the existence of distributional equivalence tests for all those circuits for which circuit equivalence is already known to be hard under complexity-theoretic assumptions. We will explore this further in Section 5.2.

**Distribution testing:** Discrete probability distributions are typically defined over an exponentially large number of points; hence a lot of recent algorithms research has focused on devising tests that require access to only a sublinear or even constant number of points in the distribution [5]. In this work, we work with distributions over $\{0,1\}^n$, and thus we aim to devise algorithms with running time at most polynomial in $n$. Previous work in testing distributions over Boolean functions has focused on the setting where the distributions offer pair-conditional sampling access [7, 22]. Using pair-conditional sampling access, Meel(r) et al. [22] were able to test distributions for closeness using $\tilde{O}(tilt(\varphi)^2/(\eta - 6\varepsilon)^3\eta)$ queries, where $tilt$ is the ratio of the probabilities of the most and least probable element in the support.

## 3  Teq: a tractable algorithm for closeness testing

In this section, we present the main contribution of the paper: a closeness test for PCs, Teq. The pseudocode of Teq is given in Algorithm 1.

Given satisfiable circuits $\varphi_1, \varphi_2$ and weight functions $\mathtt{w_1}, \mathtt{w_2}$ along with parameters $(\varepsilon, \eta, \delta)$, Teq decides whether the TV distance between $P(\varphi_1, \mathtt{w_1})$ and $P(\varphi_2, \mathtt{w_2})$ is lesser than $\varepsilon$ or greater than $\eta$ with confidence at least $1 - \delta$. Teq assumes access to an approximate weighted counter $\mathsf{Awct}(\alpha, \beta, \varphi, \mathtt{w})$, and an approximate weighted sampler $\mathsf{Samp}(\alpha, \beta, \varphi, \mathtt{w})$. We define their behavior in the following two definitions.

**The algorithm**  Teq starts by computing constants $\gamma$ and $m$. Then it queries the Awct routine with circuit $\varphi_1$ and weight function $\mathtt{w_1}$ to obtain a $\sqrt{1 + \gamma/4} - 1$ approximation of $\mathtt{w_1}(\varphi_1)$ with confidence at least $1 - \delta/8$. A similar query is made for $\varphi_2$ and $\mathtt{w_2}$ to obtain an approximate value for $\mathtt{w_2}(\varphi_2)$. These values are stored in $k_1$ and $k_2$, respectively. Teq maintains a $m$-sized array $\Gamma$, to store the estimates for $r(\sigma_i)$. Teq now iterates $m$ times. In each iteration, it generates one sample $\sigma_i$ through the Samp call on line 7. There is a small probability of at most $\delta/4m$ that this call fails and returns $\bot$. Teq only samples from one of the two PCs.

The algorithm then proceeds to compute the weight of assignment $\sigma_i$ w.r.t. the weight functions $\mathtt{w_1}$ and $\mathtt{w_2}$ and stores it in $s_1$ and $s_2$, respectively. Using the weights and approximate weighted counts stored in $k_1, k_2$ the algorithm computes the value $r(\sigma_i)$ on line 10, where $r(\sigma_i)$ is an approximation of the ratio of the probability of $\sigma_i$ in the distribution $P(\varphi_2, \mathtt{w_2})$ to its probability in $P(\varphi_1, \mathtt{w_1})$. Since $\sigma_i$ was sampled from $P(\varphi_1, \mathtt{w_1})$, its probability in $P(\varphi_1, \mathtt{w_1})$ cannot be 0, ensuring that there is no division by 0. If the ratio $r(\sigma_i)$ is less than 1, then $\Gamma[i]$ is updated with the value $1 - r(\sigma_i)$ otherwise the value of $\Gamma[i]$ remains 0. After the $m$ iterations, Teq sums up the values in the array $\Gamma$. If the sum is found to be less than threshold $m(\varepsilon + \gamma)$, Teq returns Accept and otherwise returns Reject.

The following theorem asserts the correctness of Teq.

**Theorem 1.** *Given two satisfiable probabilistic circuits $\varphi_1, \varphi_2$ and weight functions $\mathtt{w_1}, \mathtt{w_2}$, along with parameters $\varepsilon < \eta < 1$ and $\delta < 1$,*

- A. *If $d_{TV}(P(\varphi_1, \mathtt{w_1}), P(\varphi_2, \mathtt{w_2})) \leq \varepsilon$, then $\mathsf{Teq}(\varphi_1, \mathtt{w_1}, \varphi_2, \mathtt{w_2}, \varepsilon, \eta, \delta)$ returns Accept with probability at least $(1 - \delta)$.*

- B. *If $d_{TV}(P(\varphi_1, \mathtt{w_1}), P(\varphi_2, \mathtt{w_2})) \geq \eta$, then $\mathsf{Teq}(\varphi_1, \mathtt{w_1}, \varphi_2, \mathtt{w_2}, \varepsilon, \eta, \delta)$ returns Reject with probability at least $(1 - \delta)$.*

The following theorem states the running time of the algorithm,

**Theorem 2.** *Let $\gamma = \eta - \varepsilon$, then the time complexity of Teq is in* $O\left(T_{\mathsf{Awct}}(\gamma, \delta, \max(|\varphi_1|, |\varphi_2|)) + T_{\mathsf{Samp}}(\gamma, \delta, \max(|\varphi_1|, |\varphi_2|))\frac{\log(\delta^{-1})}{\gamma^2}\right)$. *If the underlying*

---
**Algorithm 1** $\mathsf{Teq}(\varphi_1, \mathtt{w}_1, \varphi_2, \mathtt{w}_2, \varepsilon, \eta, \delta)$
---
1: $\gamma \leftarrow (\eta - \varepsilon)/2$
2: $m \leftarrow \lceil 2\log(4/\delta)/\gamma^2 \rceil$
3: $\Gamma \leftarrow [0] * m$
4: $k_1 \leftarrow \mathsf{Awct}(\sqrt{1+\gamma/4} - 1, \delta/8, \varphi_1, \mathtt{w}_1)$
5: $k_2 \leftarrow \mathsf{Awct}(\sqrt{1+\gamma/4} - 1, \delta/8, \varphi_2, \mathtt{w}_2)$
6: **for** $i \in \{1, 2 \ldots, m\}$ **do**
7:     $\sigma_i \leftarrow \mathsf{Samp}(\gamma/(4\eta - 2\gamma), \delta/4m, \varphi_1, \mathtt{w}_1)$
8:     **if** $\sigma_i \neq \perp$ **then**
9:         $s_1 \leftarrow \mathtt{w}_1(\sigma_i), s_2 \leftarrow \mathtt{w}_2(\sigma_i)$
10:        $r(\sigma_i) \leftarrow \frac{s_2}{k_2} \cdot \frac{k_1}{s_1}$
11:        **if** $r(\sigma_i) < 1$ **then**
12:           $\Gamma[i] \leftarrow 1 - r(\sigma_i)$
13: **if** $\sum_{i \in [m]} \Gamma[i] \leq m(\varepsilon + \gamma)$ **then**
14:     **Return** Accept
15: **else**
16:     **Return** Reject
---

*PCs support approximate counting and sampling in polynomial time, then the running time of* $\mathsf{Teq}$ *is also polynomial in terms of* $\gamma, \log(\delta^{-1})$ *and* $max(|\varphi_1|, |\varphi_2|)$.

To improve readability, we use $P_1$ to refer to the distribution $P(\varphi_1, \mathtt{w}_1)$ and $P_2$ to refer to $P(\varphi_2, \mathtt{w}_2)$.

### 3.1 Proving the correctness of $\mathsf{Teq}$

In this subsection, we present the theoretical analysis of $\mathsf{Teq}$, and the proof of Theorem 1(A). We will defer the proofs of Theorem 1(B) and Theorem 2 to the supplementary Section A.4.2 and Section A.4.3, respectively.

For the purpose of the proof, we will first define events $\mathtt{Pass}_1, \mathtt{Pass}_2$ and $\mathtt{Good}$. Events $\mathtt{Pass}_1, \mathtt{Pass}_2$ are defined w.r.t. the function calls $\mathsf{Awct}(\sqrt{1+\gamma/4} - 1, \delta/8, \varphi_1, \mathtt{w}_1)$ and $\mathsf{Awct}(\sqrt{1+\gamma/4} - 1, \delta/8, \varphi_2, \mathtt{w}_2)$, respectively (as on lines 4, 5 of Algorithm 1). $\mathtt{Pass}_1$ and $\mathtt{Pass}_2$ represent the events that the two calls correctly return $\sqrt{1+\gamma/4}$ approximations of the weighted model counts of $\varphi_1$ and $\varphi_2$ i.e. $\frac{\mathtt{w}_1(\varphi_1)}{\sqrt{1+\gamma/4}} \leq \mathsf{Awct}(\sqrt{1+\gamma/4} - 1, \delta/8, \varphi_1, \mathtt{w}_1) \leq (\sqrt{1+\gamma/4})\mathtt{w}_1(\varphi_1)$, and $\frac{\mathtt{w}_2(\varphi_2)}{\sqrt{1+\gamma/4}} \leq \mathsf{Awct}(\sqrt{1+\gamma/4} - 1, \delta/8, \varphi_2, \mathtt{w}_2) \leq (\sqrt{1+\gamma/4})\mathtt{w}_2(\varphi_2)$. From the definition of $\mathsf{Awct}$, we have $\Pr[\mathtt{Pass}_1], \Pr[\mathtt{Pass}_2] \geq 1 - \delta/8$.

Let $\mathtt{Fail}_i$ denote the event that $\mathsf{Samp}$ (Algorithm 1, line 7) returns the symbol $\perp$ in the $i$th iteration of the loop. By the definition of $\mathsf{Samp}$ we know that $\forall_{i \in [m]} \Pr[\mathtt{Fail}_i] < \delta/4m$.

The analysis of $\mathsf{Teq}$ requires that all $m$ $\mathsf{Samp}$ calls and both $\mathsf{Awct}$ calls return correctly. We denote this super-event as $\mathtt{Good} = \bigcap_{i \in [m]} \overline{\mathtt{Fail}_i} \cap \mathtt{Pass}_1 \cap \mathtt{Pass}_2$. Applying the union bound we see that the probability of all calls to $\mathsf{Awct}$ and $\mathsf{Samp}$ returning without error is at least $1 - \delta/2$:

$$\Pr[\mathtt{Good}] = 1 - \Pr[\bigcup_{i \in [m]} \mathtt{Fail}_i \cup \overline{\mathtt{Pass}_1} \cup \overline{\mathtt{Pass}_2}] \geq 1 - m \cdot \delta/4m - 2 \cdot \delta/8 = 1 - \delta/2 \quad (1)$$

We will now state a lemma, which we will prove in the supplementary Section A.4.

**Lemma 1.** $\mathtt{Good} \rightarrow \left| r(\sigma) - \frac{P_2(\sigma)}{P_1(\sigma)} \right| \leq \gamma/4 \cdot \frac{P_2(\sigma)}{P_1(\sigma)}$

We now prove the lemma critical for our proof of correctness of $\mathsf{Teq}$.

**Lemma 2.** *Assuming the event* $\mathtt{Good}$*, let* $A = \sum_{\sigma \in \{0,1\}^n} \mathbb{1}\left(r(\sigma) < 1\right)\left(1 - r(\sigma)\right) P_1(\sigma)$*, then*

    *1. If* $d_{TV}(P_1, P_2) \leq \varepsilon$*, then* $A \leq \varepsilon + \gamma/4$

    *2. If* $d_{TV}(P_1, P_2) \geq \eta$*, then* $A \geq \eta - \gamma/4$

*Proof.* If $\sum_x (P_1(x) - P_2(x)) = 0$, then $\frac{1}{2}\sum_x |P_1(x) - P_2(x)| = \sum_{x:P_1(x)-P_2(x)>0} (P_1(x) - P_2(x))$.

Using this fact we see that,

$$d_{TV}(P_1, P_2) = \sum_{\sigma:P_2(\sigma)<P_1(\sigma)} P_1(\sigma) - P_2(\sigma) = \sum_{\sigma:\frac{P_2(\sigma)}{P_1(\sigma)}<1} \left(1 - \frac{P_2(\sigma)}{P_1(\sigma)}\right) P_1(\sigma)$$

$$= \sum_{\sigma\in\{0,1\}^n} \mathbb{1}\left(\frac{P_2(\sigma)}{P_1(\sigma)} < 1\right)\left(1 - \frac{P_2(\sigma)}{P_1(\sigma)}\right) P_1(\sigma)$$

$$= A + \underbrace{\sum_{\sigma\in\{0,1\}^n} \mathbb{1}\left(\frac{P_2(\sigma)}{P_1(\sigma)} < 1\right)\left(1 - \frac{P_2(\sigma)}{P_1(\sigma)}\right) P_1(\sigma) - A}_{B}$$

Thus we have that $d_{TV}(P_1, P_2) - A = B$. We now divide the set of assignments $\sigma \in \{0,1\}^n$ into three disjoint partition $S_1, S_2$ and $S_3$ as following: $S_1 = \{\sigma : \mathbb{1}(\frac{P_2(\sigma)}{P_1(\sigma)} < 1) = \mathbb{1}(r(\sigma) < 1)\}$; $S_2 = \{\sigma : \mathbb{1}(\frac{P_2(\sigma)}{P_1(\sigma)} < 1) > \mathbb{1}(r(\sigma) < 1)\}$; $S_3 = \{\sigma : \mathbb{1}(\frac{P_2(\sigma)}{P_1(\sigma)} < 1) < \mathbb{1}(r(\sigma) < 1)\}$. The definition implies that the indicator $\mathbb{1}(r(\sigma) < 1)$ is 0 for all assignments in the set $S_2$, and is 1 for all assignments in $S_3$. Similarly $\mathbb{1}(\frac{P_2(\sigma)}{P_1(\sigma)} < 1)$ takes value 1 and 0 for all elements in $S_2$ and $S_3$, respectively.

Now we bound the magnitude of $B$,

$$|B| = \left| \sum_{\sigma\in\{0,1\}^n} \left[\left(1 - \frac{P_2(\sigma)}{P_1(\sigma)}\right) \mathbb{1}\left(\frac{P_2(\sigma)}{P_1(\sigma)} < 1\right) - (1 - r(\sigma)) \mathbb{1}\left(r(\sigma) < 1\right)\right] P_1(\sigma) \right|$$

For $b_j > 0$, we have that $|\sum_j a_j b_j| \leq \sum_j |a_j| b_j$, and thus:

$$|B| \leq \sum_{\sigma\in\{0,1\}^n} \left| \left[\left(1 - \frac{P_2(\sigma)}{P_1(\sigma)}\right) \mathbb{1}\left(\frac{P_2(\sigma)}{P_1(\sigma)} < 1\right) - (1 - r(\sigma)) \mathbb{1}\left(r(\sigma) < 1\right)\right] \right| P_1(\sigma)$$

We can split the summation into three terms based on the sets in which the assignments lie. Some summands take the value 0 in a particular set, so we don't include them in the term.

$$|B| \leq \sum_{\sigma\in S_1} \mathbb{1}\left(\frac{P_2(\sigma)}{P_1(\sigma)} < 1\right) \left| r(\sigma) - \frac{P_2(\sigma)}{P_1(\sigma)} \right| P_1(\sigma) + \sum_{\sigma\in S_2} \mathbb{1}\left(\frac{P_2(\sigma)}{P_1(\sigma)} < 1\right)\left(1 - \frac{P_2(\sigma)}{P_1(\sigma)}\right) P_1(\sigma)$$

$$+ \sum_{\sigma\in S_3} \mathbb{1}\left(r(\sigma) < 1\right)(1 - r(\sigma)) P_1(\sigma)$$

Since we know that $\forall \sigma \in S_2, r(\sigma) > 1$ and $\forall \sigma \in S_3, \frac{P_2(\sigma)}{P_1(\sigma)} > 1$, we can alter the second and third terms of the inequality in the following way:

$$|B| \leq \sum_{\sigma\in S_1} \mathbb{1}\left(\frac{P_2(\sigma)}{P_1(\sigma)} < 1\right) \left| r(\sigma) - \frac{P_2(\sigma)}{P_1(\sigma)} \right| P_1(\sigma) + \sum_{\sigma\in S_2} \mathbb{1}\left(\frac{P_2(\sigma)}{P_1(\sigma)} < 1\right)\left| r(\sigma) - \frac{P_2(\sigma)}{P_1(\sigma)} \right| P_1(\sigma)$$

$$+ \sum_{\sigma\in S_3} \mathbb{1}\left(r(\sigma) < 1\right)\left| \frac{P_2(\sigma)}{P_1(\sigma)} - r(\sigma) \right| P_1(\sigma) \leq \sum_{\sigma\in S_1\cup S_2\cup S_3} \left| r(\sigma) - \frac{P_2(\sigma)}{P_1(\sigma)} \right| P_1(\sigma)$$

Using our assumption of the event Good and Lemma 1, $|B| \leq \sum_{\sigma\in\{0,1\}^n} \gamma/4 \cdot P_1(\sigma) \leq \gamma/4$ Since $d_{TV}(P_1, P_2) - A = B$, we get $|d_{TV}(P_1, P_2) - A| \leq \gamma/4$. We can now deduce that if $d_{TV}(P_1, P_2) \leq \varepsilon$, then $A \leq \varepsilon + \gamma/4$ and if $d_{TV}(P_1, P_2) \geq \eta$, then $A \geq \eta - \gamma/4$. $\qquad\square$

**Using** Teq **to test PCs in general.** Exact weighted model counting(WMC) is a commonly supported query on PCs. In the language of PC queries, a WMC query is known as the marginal (MAR) query. Conditional inference (CON) is another well studied PC query. Using CON and MAR, one can sample from the distribution encoded by a given PC. It is known that if a PC has the structural properties of *smoothness* and *decomposability*, then the CON and MAR queries can be computed tractably. For the definitions of the above terms and further details, please refer to the survey [9].

## 4 Evaluation

To evaluate the performance of Teq, we implemented a prototype in Python. The prototype uses WAPS[3] [17] as a weighted sampler to sample over the input d-DNNF circuits. The primary objective of our experimental evaluation was to seek an answer to the following question: Is Teq able to determine the closeness of a pair of probabilistic circuits by returning Accept if the circuits are $\varepsilon$-close and Reject if they are $\eta$-far? We test our tool Teq in the following two settings:

**A**. The pair of PCs represent small randomly generated circuits and weight functions.

**B**. The pair of PCs are from the set of publicly available benchmarks arising from sampling and counting tasks.

Our experiments were conducted on a high performance compute cluster with Intel Xeon(R) E5-2690 v3@2.60GHz CPU cores. For each benchmark, we use a single core with a timeout of 7200 seconds.

### 4.1 Setting A - Synthetic benchmarks

**Dataset** Our dataset for experiments conducted in setting **A** consisted of randomly generated 3-CNFs and with random literal weights. Our dataset consisted of 3-CNFs with $\{14, 15, 16, 17, 18\}$ variables. Since the circuits are small, we validate the results by computing the actual total variation distance using brute-force.

| Benchmark | $d_{TV}$ | | Actual | Result | Expected Result |
|:---:|:---:|:---:|:---:|:---:|:---:|
| | $\leq \epsilon$ | $\geq \eta$ | | | |
| 14_1 | 0.9 | 0.99 | 0.740 | A | A |
| 14_2 | 0.8 | 0.9 | 0.764 | A | A |
| 15_3 | 0.75 | 0.94 | 0.804 | R | A/R |
| 17_4 | 0.75 | 0.9 | 0.941 | R | R |
| 18_2 | 0.75 | 0.9 | 0.918 | R | R |

Table 1: Runtime performance of Teq. We experiment with 375 random PCs with known $d_{TV}$, and out of the 375 benchmarks we display 5 in the table and the rest in the supplementary Section B. In the table 'A' represents Accept and 'R' represents Reject. In the last column 'A/R' represents that both Accept and Reject are acceptable outputs for Teq.

**Results** Our tests terminated with the correct result in less than 10 seconds on all the randomly generated PCs we experimented with. We present the empirical results in Table 1. The first column indicates the benchmark's name, the second and third indicate the parameters $\varepsilon$ and $\eta$ on which we executed Teq. The fourth column indicates the actual $d_{TV}$ distance between the two benchmark PCs. The fifth column indicates the output of Teq, and the sixth indicates the expected result. The full detailed results are presented in the appendix Section B.

### 4.2 Setting B - Real-world benchmarks

**Dataset** We conducted experiments on a range of publicly available benchmarks arising from sampling and counting tasks[4]. Our dataset contained 100 d-DNNF circuits with weights. We have assigned random weights to literals wherever weights were not readily available. For the empirical evaluation of Teq, we needed pairs of weighted d-DNNFs with known $d_{TV}$ distance. To generate such a dataset, we first chose a circuit and a weight function, and then we synthesized new weight functions using the technique of *one variable perturbation*, described in the appendix Section B.1.

---

[3]https://github.com/meelgroup/WAPS
[4]https://zenodo.org/record/3793090

| Benchmark | $d_{TV} \leq \varepsilon$ | | $d_{TV} \geq \eta$ | |
|---|---|---|---|---|
| | Result | Teq(s) | Result | Teq(s) |
| or-70-10-8-UC-10 | A | 23.2 | R | 22.82 |
| s641_15_7 | A | 33.66 | R | 33.51 |
| or-50-5-4 | A | 414.17 | R | 408.59 |
| ProjectService3 | A | 356.15 | R | 356.14 |
| s713_15_7 | A | 24.86 | R | 24.41 |
| or-100-10-2-UC-30 | A | 31.04 | R | 31.0 |
| s1423a_3_2 | A | 153.13 | R | 152.81 |
| s1423a_7_4 | A | 104.93 | R | 103.51 |
| or-50-5-10 | A | 283.05 | R | 282.97 |
| or-60-20-6-UC-20 | A | 363.32 | R | 362.8 |

Table 2: Runtime performance of Teq. We experiment with 100 PCs with known $d_{TV}$, and out of the 100 benchmarks we display 10 in the table and the rest in the appendix B. In the table 'A' represents Accept and 'R' represents Reject. The value of the closeness parameter is $\varepsilon = 0.01$ and the farness parameter is $\eta = 0.2$.

**Results** We set the closeness parameter $\varepsilon$, farness parameter $\eta$ and confidence $\delta$ for Teq to be $0.01, 0.2$ and $0.01$, respectively. The chosen parameters imply that if the input pair of probabilistic circuits are $\leq 0.01$ close in $d_{TV}$, then Teq returns Accept with probability atleast $0.99$, otherwise if the circuits are $\geq 0.2$ far in $d_{TV}$, the algorithm returns Reject with probability at least $0.99$. The number of samples required for Teq (indicated by the variable $m$ as on line 2 of Algorithm 1) depends only on $\varepsilon, \eta, \delta$ and for the values we have chosen, we find that we require $m = 294$ samples.

Our tests terminated with the correct result in less than 3600 seconds on all the PCs we experimented with. We present the empirical results in Table 2. The first column indicates the benchmark's name, the second and third indicate the result and runtime of Teq when presented with a pair of $\varepsilon$-close PCs as input. Similarly, the fourth and fifth columns indicate the result and observed runtime of Teq when the input PCs are $\eta$-far . The full set of results are presented in the supplementary Section B.

## 5 A characterization of the complexity of testing

In this section, we characterize PCs according to the complexity of closeness and equivalence testing. We present the characterization in Table 3. The results presented in the table can be separated into (1) hardness results, and (2) upper bounds. The hardness results, presented in Section 5.2, are largely derived from known complexity-theoretic results. The upper bounds, presented in Section 5.1, are derived from a combination of established results, our algorithm Teq and the exact equivalence test of Darwiche and Huang [13](presented in supplementary Section A.1 for completeness).

### 5.1 Upper bounds

In Table 3 we label the pair of classes of PCs that admit a poly-time closeness and equivalence test with green symbols $C$ and $E$ respectively. Darwiche and Huang [13] provided an equivalence test for d-DNNF s. From Theorem 1, we know that PCs that supports the Awct and Samp queries in poly-time must also admit a poly-time approximate equivalence test. A weighted model counting algorithms for d-DNNFs was first provided by Darwiche [11], and a weighted sampler was provided by [17]. Arenas et al. [1] provided the first approximate counting and uniform sampling algorithm for SDNNFs. Using the following lemma, we show that with the use of chain formulas, the uniform sampling and counting algorithms extend to log-linear SDNNF distributions as well.

**Lemma 3.** *Given a* SDNNF *formula* $\varphi$ *(with a v-tree $T$), and a weight function* w, Samp$(\varphi, w)$ *requires polynomial time in the size of* $\varphi$.

The proof is provided in the supplementary Section A.5.

|        | NNF | PI | DNNF | SDNNF | d-DNNF |
|--------|-----|-----|------|-------|--------|
| NNF    | *EC* |    |      |       |        |
| PI     | *EC* | *UU* |     |       |        |
| DNNF   | *EC* | *EU* | *EU* |       |        |
| SDNNF  | *EC* | *EU* | *EU* | *EC*  |        |
| d-DNNF | *EC* | *UU* | *EU* | *EC*  | *EC*   |

Table 3: Summary of results. C (resp. E) indicates that a poly-time closeness (resp. equivalence) test exists. C (resp. E) indicates that a poly-time closeness (equivalence) test exists only if PH collapses. '$U$' indicates that the existence of a poly-time test is not known. The table is best viewed in color.

## 5.2 Hardness

In Table 3, we claim that the pairs of classes of PCs labeled with symbols $C$ and $E$ , cannot be tested in poly-time for closeness equivalence, respectively. Our claim assumes that the polynomial hierarchy (PH) does not collapse. To prove the hardness of testing the labeled pairs, we combine previously known facts about PCs and a few new arguments. Summarizing for brevity,

- We start off by observing that PC families are in a hierarchy, with CNF $\subseteq$ NNF and DNF $\subseteq$ SDNNF $\subseteq$ DNNF [14].
- We then reduce the problem of satisfiability testing of CNFs (NP-hard) and validity testing of DNFs (co-NP-hard) into the problem of equivalence and closeness testing of PCs, in Propositions 1, 2 and 3. These propositions and their proofs can be found in the supplementary Section A.5.
- We then connect the existence of poly-time algorithms for equivalence to the collapse of PH via a complexity result due to Karp and Lipton [18].

The NP-hardness of deciding the equivalence of pairs of DNNFs and pairs of SDNNFs was first shown by Pipatsrisawat and Darwiche [28]. We recast their proofs in the language of distribution testing for the sake of completeness in the supplementary Section A.5.

## 6 Conclusion and future work

In this paper, we studied the problem of closeness testing of PCs. Before our work, poly-time algorithms were known only for the special case of equivalence testing of PCs; and, no poly-time closeness test was known for any PC. We provided the first such test, called Teq, that used ideas from the field of distribution testing to design a novel algorithm for testing the closeness of PCs. We then implemented a prototype for Teq, and tested it on publicly available benchmarks to determine the runtime performance. Experimental results demonstrate the effectiveness of Teq in practice.

We also characterized PCs with respect to the complexity of deciding equivalence and closeness. We combined known hardness results, reductions, and our proposed algorithm Teq to classify pairs of PCs according to closeness and equivalence testing complexity. Since the characterization is incomplete, as seen in Table 3, there are questions left open regarding the existence of tests for certain PCs, which we leave for future work.

## Broader Impact

Recent advances in probabilistic modeling techniques have led to increased adoption of the said techniques in safety-critical domains, thus creating a need for appropriate verification and testing methodologies. This paper seeks to take a step in this direction and focuses on testing properties of probabilistic models likely to find use in safety-critical domains. Since our guarantees are probabilistic, practical adoption of such techniques still requires careful design to handle failures.

## Acknowledgments and Disclosure of Funding

We are grateful to the anonymous reviewers of UAI 2021 and NeurIPS 2021 for their constructive feedback that greatly improved the paper. We would also like to thank Suwei Yang and Lawqueen Kanesh for their useful comments on the earlier drafts of the paper. This work was supported in part by National Research Foundation Singapore under its NRF Fellowship Programme[NRF-NRFFAI1-2019-0004 ] and AI Singapore Programme [AISG-RP-2018-005], and NUS ODPRT Grant [R-252-000-685-13]. The computational work for this article was performed on resources of the National Supercomputing Centre, Singapore (https://www.nscc.sg).

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
