# A  Proofs omitted from the paper

## A.1  A test for equivalence

For the sake of completeness we recast the d-DNNF circuit equivalence test of Darwiche and Huang [13] into an equivalence test for log-linear probability distributions.

---

**Algorithm 2** $\text{Peq}(\varphi_1, \mathtt{w}_1, \varphi_2, \mathtt{w}_2, \delta)$

---
1: $m \leftarrow \lceil n/\delta \rceil$
2: $\theta \sim [m]^n$
3: **if** $\pi(\varphi_1, \mathtt{w}_1)(\theta) = \pi(\varphi_2, \mathtt{w}_2)(\theta)$ **then**
4:   **Return** Accept
5: **else**
6:   **Return** Reject

---

**The algorithm:**  The pseudocode for Peq is shown in Algorithm 2. Peq takes as input two satisfiable circuits $\varphi_1, \varphi_2$ defined over $n$ Boolean variables, a pair of weight functions $\mathtt{w}_1, \mathtt{w}_2$ and a tolerance parameter $\delta \in (0,1)$. Recall that a circuit $\varphi$ and a weight function $\mathtt{w}$ together define the probability distribution $P(\varphi, \mathtt{w})$. Peq returns Accept with confidence 1 if the two probability distributions $P(\varphi_1, \mathtt{w}_1)$ and $P(\varphi_2, \mathtt{w}_2)$ are equivalent, i.e. $d_{TV}(P(\varphi_1, \mathtt{w}_1), P(\varphi_2, \mathtt{w}_2)) = 0$. If $d_{TV}(P(\varphi_1, \mathtt{w}_1), P(\varphi_2, \mathtt{w}_2)) > 0$, then it returns Reject with confidence at least $1 - \delta$.

The algorithm starts by drawing a uniform random assignment $\theta$ from $[m]^n$, where $m = \lceil n/\delta \rceil$. Using the procedure given in Proposition 2 (in Section A.2), Peq computes the values $\pi(\varphi_1, \mathtt{w}_1)(\theta)$ and $\pi(\varphi_2, \mathtt{w}_2)(\theta)$, where $\pi(\varphi, \mathtt{w})$ is the network polynomial [12]. $\pi(\varphi, \mathtt{w})$ defined as:

$$\pi(\varphi, \mathtt{w}) = \sum_{\sigma \in R_\varphi} \frac{\mathtt{w}(\sigma)}{\mathtt{w}(\varphi)} \left( \prod_{x_i \models \sigma} x_i \prod_{\neg x_j \models \sigma} (1 - x_j) \right)$$

The two values are then compared on line 3, and if they are equal the algorithm returns Accept and otherwise returns Reject. The central idea of the test is that whenever the two distributions $P(\varphi_1, \mathtt{w}_1)$ and $P(\varphi_2, \mathtt{w}_2)$ are equivalent, the polynomials $\pi(\varphi_1, \mathtt{w}_1)$ and $\pi(\varphi_2, \mathtt{w}_2)$ are also equivalent, however when they are not equivalent, the polynomials disagree on atleast $1 - \delta$ fraction of assignments from the set $[m]^n$.

We formally claim and prove the correctness of Peq in Lemma 4 in the Section A.2.

## A.2  An analysis for Algorithm 2

In this section, we present the theoretical analysis of Algorithm 2 (Peq) and the proof of the following lemma.

**Lemma 4.** *Given two satisfiable probabilistic circuits $\varphi_1, \varphi_2$ and weight functions $\mathtt{w}_1, \mathtt{w}_2$, along with confidence parameter $\delta \in (0,1)$.*

- A. *If $d_{TV}(P(\varphi_1, \mathtt{w}_1), P(\varphi_2, \mathtt{w}_2)) = 0$, then $\text{Peq}(\varphi_1, \mathtt{w}_1, \varphi_2, \mathtt{w}_2, \delta)$ returns Accept with probability 1.*

- B. *If $d_{TV}(P(\varphi_1, \mathtt{w}_1), P(\varphi_2, \mathtt{w}_2)) > 0$, then $\text{Peq}(\varphi_1, \mathtt{w}_1, \varphi_2, \mathtt{w}_2, \delta)$ returns Reject with probability at least $(1 - \delta)$.*

Peq returns Accept if $\pi(\varphi_1, \mathtt{w}_1)(\sigma) = \pi(\varphi_2, \mathtt{w}_2)(\sigma)$. Since $P(\varphi_1, \mathtt{w}_1) \equiv P(\varphi_2, \mathtt{w}_2) \rightarrow \pi(\varphi_1, \mathtt{w}_1) \equiv \pi(\varphi_2, \mathtt{w}_2)$, it follows that Peq always returns Accept for two equivalent probabilistic distributions.

For the proof of Lemma 4(B) we will first define some notation, and then we show (in Lemma 5) that a random assignment over $[m]^n$ is likely to be a witness for non-equivalence with probability $> 1 - \delta$. The proof immediately follows as we know that Peq returns Reject if $\pi(\varphi_1, \mathtt{w}_1)(\sigma) \neq \pi(\varphi_2, \mathtt{w}_2)(\sigma)$.

**Definition 4.** $\pi|_{x_i=1}(\varphi, \mathtt{w})$ *is a polynomial over $n - 1$ variables, obtained by setting the variable $x_i$ to 1. Similarly $\pi|_{x_i=0}(\varphi, \mathtt{w})$ is obtained by setting the variable $x_i$ to 0, thus:*

$$\pi(\varphi, \mathtt{w}) = (1 - x_i)\pi|_{x_i=0}(\varphi, \mathtt{w}) + x_i \pi|_{x_i=1}(\varphi, \mathtt{w})$$

From the definition, we can immediately infer the following proposition.

**Proposition 1.** *If $\pi(\varphi_1, \mathtt{w_1}) \not\equiv \pi(\varphi_2, \mathtt{w_2})$ then for all $x_i$, at least one of the following must be true:*

- $\pi|_{x_i=1}(\varphi_1, \mathtt{w_1}) \neq \pi|_{x_i=1}(\varphi_2, \mathtt{w_2})$

- $\pi|_{x_i=0}(\varphi_1, \mathtt{w_1}) \neq \pi|_{x_i=0}(\varphi_2, \mathtt{w_2})$

For the proofs in this section, we will use the following notation. For a circuit $\varphi$ defined over the variables $\{x_1, \ldots, x_n\}$, we define a polynomial $P(\varphi, \mathtt{w}) : \{0,1\}^n \to [0,1]$:

$$P(\varphi, \mathtt{w}) = \sum_{\sigma \in R_\varphi} \frac{\mathtt{w}(\sigma)}{\mathtt{w}(\varphi)} \left( \prod_{x_i \models \sigma} x_i \prod_{\neg x_j \models \sigma} (1 - x_j) \right)$$

We define another polynomial $\pi(\varphi, \mathtt{w})$ which is $P(\varphi, \mathtt{w})$ but defined from $[m]^n \to \mathbb{Q}$ where $[m] = \{1 \ldots, m\}$.

To show that the polynomial $\pi(\varphi, \mathtt{w})$ can be computed in time polynomial in the size of the representation, we will adapt the procedure given by [13].

**Proposition 2.** *Let $\varphi$ be a circuit over the set $X = \{x_1, \ldots, x_n\}$ of $n$ variables , that admits poly-time WMC. Let $\mathtt{w} : X \to \mathbb{Q}^+$ be a weight function and let $\theta \in [m]^n$ be an assignment to the variables in $X$ and $\theta(x)$ be the assignment to variable $x \in X$ in $\theta$. For each node $\eta$ in the circuit, define a function $S(\cdot)$ recursively as follows:*

- $S(\eta) = \sum_i S(n_i)$, *where $\eta$ is an or-node with children $n_i$.*

- $S(\eta) = \prod_i S(n_i)$, *where $\eta$ is an and-node with children $n_i$.*

- $S(\eta) = \begin{cases} 0, & \text{if } \eta \text{ is a leaf node false} \\ 1, & \text{if } \eta \text{ is a leaf node true} \\ \mathtt{w}(x)\theta(x), & \text{if } \eta \text{ is a leaf node } x \in X \\ (1 - \mathtt{w}(x))(1 - \theta(x)), & \text{if } \eta \text{ is a leaf node } \neg x, x \in X \end{cases}$

- $\pi(\varphi, \mathtt{w}) = S(\eta)/\mathtt{w}(\varphi)$, *where $\eta$ is the root node*

*We can compute the quantity $\mathtt{w}(\varphi)$ in linear time due to our assumption of poly-time WMC, hence we can find $\pi(\varphi, \mathtt{w})(\theta)$ in time linear in the size of the* d-DNNF.

**Lemma 5.** *For a random assignment $\sigma \sim [m]^n$, $\Pr[\pi(\varphi_1, \mathtt{w_1})(\sigma) \neq \pi(\varphi_2, \mathtt{w_2})(\sigma) \mid P(\varphi_1, \mathtt{w_1}) \not\equiv P(\varphi_2, \mathtt{w_2})] > 1 - \delta$*

*Proof.* For $n = 1$, $\sigma$ is an assignment to a single variable $x$. The polynomial on the single variable $x$ can be parameterised as $\pi(\varphi, \mathtt{w})(x) = \alpha x + (1 - \alpha)(1 - x)$ where parameter $\alpha = \frac{\mathtt{w}(\mathtt{x})}{\sum_{\theta \in R_\varphi} \mathtt{w}(\theta)}$.

Let polynomials $\pi(\varphi_1, \mathtt{w_1}), \pi(\varphi_2, \mathtt{w_2})$ be parameterised with $\alpha_1, \alpha_2$, respectively. Our assumption that $P(\varphi_1, \mathtt{w_1}) \not\equiv P(\varphi_2, \mathtt{w_2})$ immediately leads to the fact that $\pi(\varphi_1, \mathtt{w_1}) \not\equiv \pi(\varphi_2, \mathtt{w_2})$ which in turn implies that $\alpha_1 \neq \alpha_2$.

The the set of inputs $x$ for which two non-equivalent polynomials agree is given by,

$$\pi(\varphi_1, \mathtt{w_1})(x) = \pi(\varphi_2, \mathtt{w_2})(x)$$
$$\alpha_1 x + (1 - \alpha_1)(1 - x) = \alpha_2 x + (1 - \alpha_2)(1 - x)$$
$$2(\alpha_1 - \alpha_2)x = \alpha_1 - \alpha_2$$
$$x = 1/2$$

From the initial assumption we know that $x$ can only take integer values, hence there are no inputs in the set $[m]$ for which $\pi(\varphi_1, \mathtt{w_1})(\sigma) \neq \pi(\varphi_2, \mathtt{w_2})(\sigma)$. Thus, for $n = 1$, and any $\sigma$, $\Pr[\pi(\varphi_1, \mathtt{w_1})(\sigma) \neq \pi(\varphi_2, \mathtt{w_2})(\sigma) \mid P(\varphi_1, \mathtt{w_1}) \not\equiv P(\varphi_2, \mathtt{w_2})] = 0$

We now assume that the hypothesis holds for $n - 1$ variables. Consider polynomials $\pi(\varphi_1, \mathtt{w_1}) \not\equiv \pi(\varphi_2, \mathtt{w_2})$ over $n$ variables. From Prop 1 we know that at least one of the following holds:

- $\pi|_{x_i=1}(\varphi_1, \mathtt{w_1}) \neq \pi|_{x_i=1}(\varphi_2, \mathtt{w_2})$

- $\pi|_{x_i=0}(\varphi_1, \mathtt{w_1}) \neq \pi|_{x_i=0}(\varphi_2, \mathtt{w_2})$

Without any loss of generality we assume the latter. Then we know that there exists a set $\Sigma \subseteq [m]^{n-1}, |\Sigma| \geq (m-1)^{n-1}$, such that

$$\forall_{\sigma \in \Sigma}, \pi|_{x_n=0}(\varphi_1, \mathtt{w_1})(\sigma) \neq \pi|_{x_n=0}(\varphi_2, \mathtt{w_2})(\sigma)$$

The set of assignments $\sigma$ for which $\pi(\varphi_1, \mathtt{w_1})(\sigma) = \pi(\varphi_2, \mathtt{w_2})(\sigma)$ is given by,

$$\pi(\varphi_1, \mathtt{w_1})(\sigma) = \pi(\varphi_2, \mathtt{w_2})(\sigma)$$

$$(1 - x_n)\pi|_{x_n=0}(\varphi_1, \mathtt{w_1})(\sigma) + x_n\pi|_{x_n=1}(\varphi_1, \mathtt{w_1})(\sigma) = (1 - x_n)\pi|_{x_n=0}(\varphi_2, \mathtt{w_2})(\sigma) + x_n\pi|_{x_n=1}(\varphi_2, \mathtt{w_2})(\sigma)$$

$$x_n(\pi|_{x_n=1}(\varphi_1, \mathtt{w_1})(\sigma) - \pi|_{x_n=0}(\varphi_1, \mathtt{w_1})(\sigma) - \pi|_{x_n=1}(\varphi_2, \mathtt{w_2})(\sigma) + \pi|_{x_n=0}(\varphi_2, \mathtt{w_2})(\sigma))$$
$$= \pi|_{x_n=0}(\varphi_2, \mathtt{w_2})(\sigma) - \pi|_{x_n=0}(\varphi_1, \mathtt{w_1})(\sigma)$$

From the assumptions we know that there are at least $(m-1)^{n-1}$ assignments $\sigma$ s.t. $\pi|_{x_n=0}(\varphi_2, \mathtt{w_2})(\sigma) - \pi|_{x_n=0}(\varphi_1, \mathtt{w_1})(\sigma) \neq 0$, from which we can conclude that the RHS is non-zero. Thus for all such $\sigma$ there can be at most one value of $x_n$ for which the equality holds, which leaves $m-1$ values which $x_n$ cannot take. Thus there are at least $(m-1) \times (m-1)^{n-1} = (m-1)^n$ assignments to $n$ variables for which $\pi(\varphi_1, \mathtt{w_1})(\sigma) \neq \pi(\varphi_2, \mathtt{w_2})(\sigma)$.

Since the total number of assignments for $n$ variables is $m^n$, out of which $(m-1)^n$ witness the non-equivalence of the two probability distributions, we know that for a randomly chosen assignment $\sigma \sim [m]^n$, we have

$$\Pr[\pi(\varphi_1, \mathtt{w_1})(\sigma) \neq \pi(\varphi_2, \mathtt{w_2})(\sigma) \mid P(\varphi_1, \mathtt{w_1}) \not\equiv P(\varphi_2, \mathtt{w_2})] \geq \frac{(m-1)^n}{m^n} \geq \left(1 - \frac{\delta}{n}\right)^n$$

$$> 1 - \delta \quad \text{(using } m \text{ from Algorithm 2)}$$

$\square$

## A.3 Omitted proof from the analysis of Algorithm 1

In this subsection, we present the proof of Theorem 1(B), and Theorem 2. Recall that we use $P_1$ and $P_2$ to refer to $P(\varphi_1, \mathtt{w_1})$ and $P(\varphi_2, \mathtt{w_2})$, respectively.

## A.4 Proof of Lemma 1

*Proof.* The quantity $r(\sigma)$ (line 10 from Algorithm 1) conditioned on the event $\overline{\mathtt{Fail_i}} \subset \mathtt{Good}$:

$$r(\sigma) = \frac{\mathtt{w_2}(\sigma)}{\mathsf{Awct}(\sqrt{1+\gamma/4}-1, \delta/8, \varphi_2, \mathtt{w_2})} \cdot \frac{\mathsf{Awct}(\sqrt{1+\gamma/4}-1, \delta/8, \varphi_1, \mathtt{w_1})}{\mathtt{w_1}(\sigma)}$$

Conditioned on the events $\mathtt{Pass_1}, \mathtt{Pass_2} \subset \mathtt{Good}$, we know that with probability 1:

$$\frac{\mathtt{w_2}(\sigma)\mathtt{w_1}(\varphi_1)}{\mathtt{w_2}(\varphi_2)\mathtt{w_1}(\sigma)}(\sqrt{1+\gamma/4})^{-2} < r(\sigma) < (\sqrt{1+\gamma/4})^2 \frac{\mathtt{w_2}(\sigma)\mathtt{w_1}(\varphi_1)}{\mathtt{w_2}(\varphi_2)\mathtt{w_1}(\sigma)}$$

Which gives us: $\frac{P_2(\sigma)}{P_1(\sigma)}(1+\gamma/4)^{-1} < r(\sigma) < (1+\gamma/4)\frac{P_2(\sigma)}{P_1(\sigma)}$ and therefore,

$$\left| r(\sigma) - \frac{P_2(\sigma)}{P_1(\sigma)} \right| \le \frac{P_2(\sigma)}{P_1(\sigma)} \cdot \max_{0<\gamma<1}\left(\gamma/4, 1-\frac{1}{1+\gamma/4}\right) \le \frac{P_2(\sigma)}{P_1(\sigma)} \cdot \gamma/4 \qquad \square$$

### A.4.1 Proof of Theorem 1(A)

*Proof.* We assume the event $\mathtt{Good}$. Let $\sigma_i$ be the sample returned by the sampler $\mathsf{Samp}$ in the $i$th iteration. If $r(\sigma_i) > 1$, $\Gamma[i]$ takes value 0, else $\Gamma[i] = 1 - r(\sigma_i)$. Thus $\Gamma[i]$ is a r.v. which takes on a value from $[0,1]$. We can write $\Gamma[i] = \mathbb{1}\left(r(\sigma_i) < 1\right)\left(1-r(\sigma_i)\right)$ The expectation of $\Gamma[i]$ is:

$$\mathbb{E}[\Gamma[i]] = \sum_{\sigma \in \{0,1\}^n} \mathbb{1}\left(r(\sigma) < 1\right)\left(1-r(\sigma)\right) \cdot \Pr[\mathsf{Samp}(\gamma/(4\eta-2\gamma), \delta/4m, \varphi_1, \mathtt{w_1}) = \sigma] \qquad (2)$$

According to definition 3, and our assumption of $\overline{\mathtt{Fail_i}} \subset \mathtt{Good}$, we know that with probability 1 $\Pr[\mathsf{Samp}(\gamma/(4\eta-2\gamma), \delta/4m, \varphi_1, \mathtt{w_1}) = \sigma] \le (1+\gamma/(4\eta-2\gamma))P_1(\sigma)$. Thus we have,

$$\mathbb{E}[\Gamma[i]] \le \sum_{\sigma \in \{0,1\}^n} \mathbb{1}\left(r(\sigma) < 1\right)\left(1-r(\sigma)\right) \cdot (1+\gamma/(4\eta-2\gamma))P_1(\sigma)$$

Recall that in Lemma 2, we define $A = \sum_{\sigma \in \{0,1\}^n} \mathbb{1}\left(r(\sigma) < 1\right)\left(1-r(\sigma)\right)P_1(\sigma)$. Therefore, we can simplify the above expression as: $\mathbb{E}[\Gamma[i]] = (1+\gamma/(4\eta-2\gamma))\cdot A$. We can then use the assumption of $\varepsilon$-closeness and the result of Lemma 2-1 to find a bound on the expectation,

$$\mathbb{E}[\Gamma[i]] \le (1+\gamma/(4\eta-2\gamma))(\varepsilon+\gamma/4) \le \varepsilon + \gamma/2$$

Using the linearity of expectation we get: $\mathbb{E}\left[\sum_{i\in[m]}\Gamma[i]\right] < m(\varepsilon+\gamma/2)$. Teq returns Reject when $\sum_{i\in[m]}\Gamma[i] > m(\varepsilon+\gamma)$ on line 13. Since the $\Gamma[i]$'s are i.i.d random variables taking values in $[0,1]$, we apply the Chernoff bound to find the probability of Accept, assuming the event $\mathtt{Good}$:

$$\Pr\left[\mathsf{Teq} \text{ returns Accept} \,\middle|\, \mathtt{Good}\right] = 1 - \Pr\left[\sum_{i\in[m]}\Gamma[i] > m(\varepsilon+\gamma)\right] \ge 1 - 2e^{-\gamma^2 m/2} \ge 1 - \delta/2$$

The value for $m$ is taken from line 2 of Algorithm 1. Using (1), we see that the probability of Teq returning Accept is: $\Pr[\mathsf{Teq} \text{ returns Accept}] \ge \Pr[\mathsf{Teq} \text{ returns Accept} \mid \mathtt{Good}]\Pr[\mathtt{Good}] = (1-\delta/2)(1-\delta/2) \ge 1-\delta$ $\qquad \square$

### A.4.2 Proof of Theorem 1(B)

*Proof.* First we assume the event $\mathtt{Good}$. Then according to definition 3, we know that with probability 1 (since we assume event $\overline{\mathtt{Fail_i}} \subset \mathtt{Good}$)

$$\Pr[\mathsf{Samp}(\gamma/(4\eta-2\gamma), \delta/4m, \varphi_1, \mathtt{w_1}) = \sigma] \ge \frac{P_1(\sigma)}{(1+\gamma/(4\eta-2\gamma))}$$

Thus substituting into (2), we get

$$\mathbb{E}[\Gamma[i]] \geq \sum_{\sigma \in \{0,1\}^n} \mathbb{1}\left(r(\sigma) < 1\right)(1 - r(\sigma)) \frac{P_1(\sigma_i)}{1 + \gamma/(4\eta - 2\gamma)} \tag{3}$$

Then we use the $\eta$-farness assumption and Lemma 2-2

$$\mathbb{E}[\Gamma[i]] \geq \frac{\eta - \gamma/4}{1 + \gamma/(4\eta - 2\gamma)} = \eta - \gamma/2 \tag{4}$$

The algorithm returns Accept when $\sum_{i \in [m]} \Gamma[i] \leq m(\varepsilon + \gamma)$ (on line 13). Then using (4) and the linearity of expectation.

$$\mathbb{E}\left[\sum_{i \in [m]} \Gamma[i]]\right] \geq m(\eta - \gamma/2)$$

Since the $\Gamma[i]$'s are i.i.d random variables taking values in $[0, 1]$, we apply the Chernoff bound to find the probability of Reject, given the assumption of the event Good:

$$
\begin{aligned}
\Pr\left[\text{Teq returns Reject} \mid \text{Good}\right] &= 1 - \Pr\left[\sum_{i \in [m]} \Gamma[i] \leq m(\varepsilon + \gamma)\right] \\
&\geq 1 - \Pr\left[m(\eta - \gamma/2) - \sum_{i \in [m]} \Gamma[i] \geq m(\eta - \gamma/2 - \varepsilon - \gamma)\right] \\
&\geq 1 - \Pr\left[|\sum_{i \in [m]} \Gamma[i] - m(\eta - \gamma/2)| \geq m\gamma/2\right] \\
&\geq 1 - 2e^{-\gamma^2 m/2} \geq 1 - \delta/2 \quad \text{(Substituting } m \text{ as in line 2)}
\end{aligned}
$$

Hence, the probability that Algorithm 1 returns Reject is

$$
\begin{aligned}
\Pr[\text{Teq returns Reject}] &\geq \Pr\left[\text{Teq returns Reject} \mid \text{Good}\right] \Pr\left[\text{Good}\right] \\
&= (1 - \delta/2)(1 - \delta/2) \geq 1 - \delta \quad \text{(Using (1))}
\end{aligned}
$$

$\square$

### A.4.3 Proof of Theorem 2

*Proof.* Teq makes two calls to Awct on line 4 and 5 of Algorithm 1. According to definition 2, the runtime of the $\text{Awct}(\sqrt{1 + \gamma/4} - 1, \delta/8, \varphi, \mathtt{w})$ query is $T(\sqrt{1 + \gamma/4} - 1, \delta/8, \varphi) = poly((\sqrt{1 + \gamma/4} - 1)^{-1}, \log(\delta^{-1}), |\varphi|)$.

Using the identity $1 + \frac{x}{2} - \frac{x^2}{2} \leq \sqrt{1 + x}$ for $x \geq 0$ and the fact that $\gamma \in (0, 1)$

$$\frac{1}{\sqrt{1 + \gamma/4} - 1} \leq \frac{1}{\gamma/8 - \gamma^2/32} < \frac{11}{\gamma}$$

Hence any $poly((\sqrt{1 + \gamma/4} - 1)^{-1})$ algorithm also runs in $poly(\gamma^{-1})$. Thus the Awct queries run in $O(poly(\gamma^{-1}, \log(\delta^{-1}), max(|\varphi_1|, |\varphi_2|)))$

Teq makes $m = \lceil \log(2/\delta)/2\gamma^2 \rceil$ calls to Samp on lines 7 of Algorithm 1. According to definition 3, the runtime of the $\text{Samp}(\gamma/(4\eta - 2\gamma), \delta/4m, \varphi_1, \mathtt{w_1})$ query is $T(\gamma/(4\eta - 2\gamma), \delta/4m, |\varphi_1|) = poly((\gamma/(4\eta - 2\gamma))^{-1}, \log((\delta/4m)^{-1}), |\varphi_1|)$. First we see that $\frac{4\eta - 2\gamma}{\gamma} < \frac{4}{\gamma}$, thus the algorithm remains in $poly(\gamma^{-1})$. We then see that $\log(4m/\delta) = \log(4m) + \log(\delta^{-1})$. Since $\log(m) \in poly(\log(\gamma^{-1}), \log\log(\delta^{-1}))$, we know that Samp queries run in $O(poly(\gamma^{-1}, \log(\delta^{-1}), max(|\varphi_1|, |\varphi_2|)))$.

Since each Samp call and each Awct call requires atmost polynomial time in terms of $\gamma^{-1}, \log(\delta^{-1})$ and $max(|\varphi_1|, |\varphi_2|)$ we know that the algorithm itself runs in time polynomial in $\gamma^{-1}, \log(\delta^{-1})$ and $max(|\varphi_1|, |\varphi_2|)$. $\square$

### A.5 Proofs omitted from Section 5

For the following proofs, we assume a uniform weight function.

**Proposition 1.** *If there exists a poly-time randomised algorithm for deciding the equivalence of a pair of PCs with at least one PC in* CNF*, then NP=RP.*

*Proof.* For CNFs, testing satisfiability is known to be NP-hard. Consider a CNF $\varphi$ defined over variables $\{x_1, \ldots, x_n\}$ and a circuit $\psi$ s.t. $\psi \equiv \bigwedge_{i \in [n+1]} x_i$. Define

$$\hat{\varphi} = (\neg x_{n+1} \rightarrow \varphi) \wedge (x_{n+1} \rightarrow \bigwedge_{i \in [n]} x_i)$$

We see that the size of the new CNF is $|\hat{\varphi}| \in O(|\varphi| + n)$. $\hat{\varphi}$ has at least one satisfying assignment, specifically the assignment $\forall_{i \in [n+1]} x_i = 1$. We notice that $d_{TV}(P(\hat{\varphi}, \mathtt{w}), P(\psi, \mathtt{w})) = 0$ if and only if $|R_\varphi| = 0$. Thus the existence of a poly-time randomised algorithm for deciding whether $d_{TV}(P(\hat{\varphi}, \mathtt{w}), P(\psi, \mathtt{w})) = 0$ would imply NP $\subseteq$ RP and hence NP=RP. $\square$

**Proposition 2.** *If there exists a poly-time randomised algorithm for deciding the closeness of a pair of PCs with at least one PC in* CNF*, then NP=RP.*

*Proof.* $d_{TV}(P(\hat{\varphi}, \mathtt{w}), P(\psi, \mathtt{w})) \geq 0.5$ if and only if $|R_\varphi| > 0$. Assume there exists a poly-time randomised algorithm which returns Reject if $d_{TV}(P(\hat{\varphi}, \mathtt{w}), P(\psi, \mathtt{w})) \geq 0.4$ and Accept if $d_{TV}(P(\hat{\varphi}, \mathtt{w}), P(\psi, \mathtt{w})) \leq 0.1$ with probability $> 2/3$. Such an algorithm would imply BPP $\subseteq$ NP, and hence NP=RP. $\square$

**Proposition 3.** *If there exists a poly-time randomised algorithm for deciding the equivalence of a pair of PCs with at least one PC in* DNF*, then NP=RP.*

*Proof.* For DNFs, deciding validity is known to be co-NP-hard. Given DNF $\varphi$ and a circuit $\psi = True$, the existence of a poly-time randomized algorithm for checking the equivalence of $\psi$ and $\varphi$ would imply that co-NP $\subseteq$ co-RP and hence co-NP = co-RP. $\square$

Using Corollary 6.3 from [18] we see that PH collapses as a result of either of the above implications.

From the set inclusions DNF$\subseteq$ SDNNF$\subseteq$ DNNF and CNF$\subseteq$ NNF, we obtain all hardness results. From the fact that d-DNNFs support weighted counting and sampling we have the existence results.

The following lemma supports our claim in table 3.

**Lemma 6.** *Given a* SDNNF *formula $\varphi$ (with a v-tree $T$), and a weight function $\mathtt{w}$,* $\mathsf{Samp}(\varphi, \mathtt{w})$ *requires polynomial time in the size of $\varphi$.*

*Proof.* Here we will assume that the weights are in the dyadic form i.e. they can be represented as the fraction $d/2^p$ for $d, p \in \mathbb{Z}^+$. Then using the weighted to unweighted construction from [8], the problem of approximate weighted sampling over SDNNF can be reduced to approximate uniform sampling. Given a SDNNF $\varphi$, and a weight function $\mathtt{w}$, we generate a SDNNF $\varphi_w \equiv \varphi \wedge \bigwedge_{i \in [n]} (\neg x_i \vee C_i^1) \wedge \bigwedge_{i \in [n]} (x_i \vee C_i^0)$. Here, $C_i^0$ is chain formula having exactly $w(\neg x_i) \times 2^p = 2^p - d$ satisfying assignments, and $C_i^1$ is a chain formula with $\mathtt{w}(x_i) \times 2^p = d$ satisfying assignments.

The property of decomposability on the $\wedge$ nodes of $\varphi$ is preserved as each $C_i$ introduces a new set of variables disjoint from the set of variables in $\varphi$ and and also from all $C_j$, such that $j \neq i$. The $\wedge$ nodes in the chain formula are also trivially decomposable and structured as each chain formula variable appears exactly once in the formula.

If $\sigma$ is an assignment to the set of variables of $S$ and if $S' \subseteq S$, then let $\sigma_{\downarrow S'}$ denote the projection of $\sigma$ on the variables in $S'$. The weighted formula $\varphi$ is defined over variable set $var(\varphi)$. The formula $\varphi_w$ defined above has the property that if $\varphi(\sigma) = 1$, then $|\{\sigma'|\varphi_w(\sigma') = 1 \wedge \sigma'_{\downarrow var(\varphi)} = \sigma\}|/|R_{\varphi_w}| = \mathtt{w}(\sigma)$. Thus a uniform distribution on $R_{\varphi_w}$, when projected on $var(\varphi)$ induces the weighted distribution $P(\varphi, \mathtt{w})$. This property allows weighted sampling and counting on $\varphi$ with the help of a uniform sampler for the generated formula $\varphi_w$.

$\square$

# B Experimental evaluation

In this section we will first discuss the method for generating the synthetic dataset, and then we present the extended table of results.

## B.1 One variable perturbation

Consider two weight functions $\mathtt{w}_1$ and $\mathtt{w}_2$ that differ only in the weight assigned to the literals $v^0$ and $v^1$. Then, from the definition of $d_{TV}$:

$$d_{TV}(P(\varphi, \mathtt{w}_1), P(\varphi, \mathtt{w}_2)) = \frac{1}{2} \sum_{\sigma \in \{0,1\}^n} \left| \frac{\mathtt{w}_1(\sigma)}{\mathtt{w}_1(\varphi)} - \frac{\mathtt{w}_2(\sigma)}{\mathtt{w}_2(\varphi)} \right|$$

Let $S \subseteq \{0,1\}^n$ be the set of assignments for which $\frac{\mathtt{w}_1(\sigma)}{\mathtt{w}_1(\varphi)} > \frac{\mathtt{w}_2(\sigma)}{\mathtt{w}_2(\varphi)}$. Thus,

$$d_{TV}(P(\varphi, \mathtt{w}_1), P(\varphi, \mathtt{w}_2)) = \sum_{\sigma \in S} \left( \frac{\mathtt{w}_1(\sigma)}{\mathtt{w}_1(\varphi)} - \frac{\mathtt{w}_2(\sigma)}{\mathtt{w}_2(\varphi)} \right)$$

Lets assume wlog that $\mathtt{w}_1$ assigns a larger weight to $v^1$ than $\mathtt{w}_2$ does. Then, $S$ contains all and only those assignments that have literal $v^1$, i.e. $S \equiv \varphi \wedge v^1$. Thus,

$$d_{TV}(P(\varphi, \mathtt{w}_1), P(\varphi, \mathtt{w}_2)) = \frac{\mathtt{w}_1(\varphi \wedge v^1)}{\mathtt{w}_1(\varphi)} - \frac{\mathtt{w}_2(\varphi \wedge v^1)}{\mathtt{w}_2(\varphi)}$$

We can rewrite $\mathtt{w}_1(\varphi \wedge v^1) = \mathtt{w}_1'(\varphi) \times \mathtt{w}_1(v^1)$, where $\mathtt{w}_1'$ is $\mathtt{w}_1$ with the weight of $v^1$ set to 1. Using a similar transformation on $\mathtt{w}_2(\varphi \wedge v^1)$ we get

$$d_{TV}(P(\varphi, \mathtt{w}_1), P(\varphi, \mathtt{w}_2)) = \frac{\mathtt{w}_1'(\varphi) \times \mathtt{w}_1(v^1)}{\mathtt{w}_1(\varphi)} - \frac{\mathtt{w}_2'(\varphi) \times \mathtt{w}_2(v^1)}{\mathtt{w}_2(\varphi)}$$

We know that $\mathtt{w}_1'(\varphi) = \mathtt{w}_2'(\varphi)$ as $\mathtt{w}_1$ and $\mathtt{w}_2$ differed only on the one variable $v^1$.

$$d_{TV}(P(\varphi, \mathtt{w}_1), P(\varphi, \mathtt{w}_2)) = \mathtt{w}_1'(\varphi) \times \left( \frac{\mathtt{w}_1(v^1)}{\mathtt{w}_1(\varphi)} - \frac{\mathtt{w}_2(v^1)}{\mathtt{w}_2(\varphi)} \right)$$

All quantities in the above expression are either known constants or they are defined w.r.t the already compiled d-DNNF, thus guaranteeing that $d_{TV}(P(\varphi, \mathtt{w}_1), P(\varphi, \mathtt{w}_2))$ can be computed in poly-time.

## B.2 Extended table of results

The timeout for all our experiments was set to 7200 seconds.

### B.2.1 Synthetic PCs

In the following table, the first column indicates the benchmark, the second and the third indicate the closeness parameter $\varepsilon$ and $\eta$ used in the test. The fourth column indicates actual $d_{TV}$ distance between the two benchmark PCs . The fifth column indicates the test outcome and the sixth represents the expected outcome. 'A' represents Accept and 'R' represents Reject and 'A/R' represents that both 'A' and 'R' are acceptable outputs.

Table 4: The Extended Table

| Benchmark | $\varepsilon$ | $\eta$ | Actual $d_{TV}$ | Result | Expected Result |
|---|---|---|---|---|---|
| 14_1 | 0.75 | 0.99 | 0.740 | A | A |
| 14_1 | 0.85 | 0.94 | 0.740 | A | A |
| 14_1 | 0.75 | 0.96 | 0.740 | A | A |

| 14_1 | 0.9 | 0.94 | 0.740 | A | A |
|---|---|---|---|---|---|
| 14_1 | 0.85 | 0.9 | 0.740 | A | A |
| 14_1 | 0.8 | 0.96 | 0.740 | A | A |
| 14_1 | 0.75 | 0.94 | 0.740 | A | A |
| 14_1 | 0.8 | 0.9 | 0.740 | A | A |
| 14_1 | 0.9 | 0.96 | 0.740 | A | A |
| 14_1 | 0.85 | 0.99 | 0.740 | A | A |
| 14_1 | 0.8 | 0.94 | 0.740 | A | A |
| 14_1 | 0.8 | 0.99 | 0.740 | A | A |
| 14_1 | 0.85 | 0.96 | 0.740 | A | A |
| 14_1 | 0.75 | 0.9 | 0.740 | A | A |
| 14_1 | 0.9 | 0.99 | 0.740 | A | A |
| 14_2 | 0.9 | 0.99 | 0.764 | A | A |
| 14_2 | 0.85 | 0.94 | 0.764 | A | A |
| 14_2 | 0.9 | 0.96 | 0.764 | A | A |
| 14_2 | 0.85 | 0.99 | 0.764 | A | A |
| 14_2 | 0.75 | 0.96 | 0.764 | A | A/R |
| 14_2 | 0.8 | 0.94 | 0.764 | A | A |
| 14_2 | 0.75 | 0.94 | 0.764 | A | A/R |
| 14_2 | 0.75 | 0.9 | 0.764 | A | A/R |
| 14_2 | 0.8 | 0.9 | 0.764 | A | A |
| 14_2 | 0.75 | 0.99 | 0.764 | A | A/R |
| 14_2 | 0.85 | 0.9 | 0.764 | A | A |
| 14_2 | 0.8 | 0.99 | 0.764 | A | A |
| 14_2 | 0.9 | 0.94 | 0.764 | A | A |
| 14_2 | 0.8 | 0.96 | 0.764 | A | A |
| 14_2 | 0.85 | 0.96 | 0.764 | A | A |
| 14_0 | 0.75 | 0.96 | 0.771 | A | A/R |
| 14_0 | 0.9 | 0.94 | 0.771 | A | A |
| 14_0 | 0.75 | 0.99 | 0.771 | A | A/R |
| 14_0 | 0.8 | 0.96 | 0.771 | A | A |
| 14_0 | 0.85 | 0.94 | 0.771 | A | A |
| 14_0 | 0.85 | 0.9 | 0.771 | A | A |
| 14_0 | 0.75 | 0.94 | 0.771 | A | A/R |
| 14_0 | 0.85 | 0.96 | 0.771 | A | A |
| 14_0 | 0.8 | 0.9 | 0.771 | A | A |
| 14_0 | 0.9 | 0.99 | 0.771 | A | A |
| 14_0 | 0.9 | 0.96 | 0.771 | A | A |
| 14_0 | 0.85 | 0.99 | 0.771 | A | A |
| 14_0 | 0.8 | 0.94 | 0.771 | A | A |

| 14_0 | 0.8 | 0.99 | 0.771 | A | A |
|------|------|------|-------|---|-----|
| 14_0 | 0.75 | 0.9 | 0.771 | A | A/R |
| 14_4 | 0.9 | 0.99 | 0.773 | A | A |
| 14_4 | 0.85 | 0.94 | 0.773 | A | A |
| 14_4 | 0.9 | 0.96 | 0.773 | A | A |
| 14_4 | 0.85 | 0.99 | 0.773 | A | A |
| 14_4 | 0.75 | 0.96 | 0.773 | A | A/R |
| 14_4 | 0.8 | 0.94 | 0.773 | A | A |
| 14_4 | 0.8 | 0.9 | 0.773 | A | A |
| 14_4 | 0.8 | 0.99 | 0.773 | A | A |
| 14_4 | 0.75 | 0.9 | 0.773 | A | A/R |
| 14_4 | 0.9 | 0.94 | 0.773 | A | A |
| 14_4 | 0.85 | 0.9 | 0.773 | A | A |
| 14_4 | 0.75 | 0.99 | 0.773 | A | A/R |
| 14_4 | 0.8 | 0.96 | 0.773 | A | A |
| 14_4 | 0.75 | 0.94 | 0.773 | A | A/R |
| 14_4 | 0.85 | 0.96 | 0.773 | A | A |
| 15_3 | 0.75 | 0.99 | 0.804 | R | A/R |
| 15_3 | 0.8 | 0.99 | 0.804 | R | A/R |
| 15_3 | 0.9 | 0.94 | 0.804 | A | A |
| 15_3 | 0.8 | 0.96 | 0.804 | R | A/R |
| 15_3 | 0.85 | 0.96 | 0.804 | A | A |
| 15_3 | 0.9 | 0.99 | 0.804 | A | A |
| 15_3 | 0.85 | 0.94 | 0.804 | A | A |
| 15_3 | 0.85 | 0.9 | 0.804 | R | A/R |
| 15_3 | 0.9 | 0.96 | 0.804 | A | A |
| 15_3 | 0.85 | 0.99 | 0.804 | A | A |
| 15_3 | 0.8 | 0.94 | 0.804 | R | A/R |
| 15_3 | 0.75 | 0.94 | 0.804 | R | A/R |
| 15_3 | 0.75 | 0.96 | 0.804 | R | A/R |
| 15_3 | 0.75 | 0.9 | 0.804 | R | A/R |
| 15_3 | 0.8 | 0.9 | 0.804 | R | A/R |
| 16_4 | 0.75 | 0.96 | 0.833 | A | A/R |
| 16_4 | 0.75 | 0.9 | 0.833 | A | A/R |
| 16_4 | 0.9 | 0.96 | 0.833 | A | A |
| 16_4 | 0.85 | 0.96 | 0.833 | A | A |
| 16_4 | 0.8 | 0.9 | 0.833 | A | A/R |
| 16_4 | 0.9 | 0.99 | 0.833 | A | A |
| 16_4 | 0.75 | 0.99 | 0.833 | A | A/R |
| 16_4 | 0.8 | 0.94 | 0.833 | A | A/R |

| 16_4 | 0.85 | 0.99 | 0.833 | A | A |
|------|------|------|-------|---|-----|
| 16_4 | 0.75 | 0.94 | 0.833 | A | A/R |
| 16_4 | 0.8 | 0.99 | 0.833 | A | A/R |
| 16_4 | 0.9 | 0.94 | 0.833 | A | A |
| 16_4 | 0.85 | 0.94 | 0.833 | A | A |
| 16_4 | 0.8 | 0.96 | 0.833 | A | A/R |
| 16_4 | 0.85 | 0.9 | 0.833 | A | A |
| 14_3 | 0.8 | 0.96 | 0.852 | A | A/R |
| 14_3 | 0.8 | 0.94 | 0.852 | A | A/R |
| 14_3 | 0.75 | 0.94 | 0.852 | A | A/R |
| 14_3 | 0.85 | 0.96 | 0.852 | A | A/R |
| 14_3 | 0.75 | 0.9 | 0.852 | R | A/R |
| 14_3 | 0.8 | 0.9 | 0.852 | R | A/R |
| 14_3 | 0.9 | 0.99 | 0.852 | A | A |
| 14_3 | 0.85 | 0.9 | 0.852 | A | A/R |
| 14_3 | 0.75 | 0.99 | 0.852 | A | A/R |
| 14_3 | 0.85 | 0.99 | 0.852 | A | A/R |
| 14_3 | 0.8 | 0.99 | 0.852 | A | A/R |
| 14_3 | 0.75 | 0.96 | 0.852 | A | A/R |
| 14_3 | 0.85 | 0.94 | 0.852 | A | A/R |
| 14_3 | 0.9 | 0.94 | 0.852 | A | A |
| 14_3 | 0.9 | 0.96 | 0.852 | A | A |
| 17_1 | 0.8 | 0.96 | 0.874 | R | A/R |
| 17_1 | 0.85 | 0.94 | 0.874 | A | A/R |
| 17_1 | 0.75 | 0.96 | 0.874 | R | A/R |
| 17_1 | 0.85 | 0.99 | 0.874 | A | A/R |
| 17_1 | 0.75 | 0.9 | 0.874 | R | A/R |
| 17_1 | 0.75 | 0.94 | 0.874 | R | A/R |
| 17_1 | 0.8 | 0.9 | 0.874 | R | A/R |
| 17_1 | 0.8 | 0.94 | 0.874 | R | A/R |
| 17_1 | 0.75 | 0.99 | 0.874 | R | A/R |
| 17_1 | 0.9 | 0.94 | 0.874 | A | A |
| 17_1 | 0.85 | 0.9 | 0.874 | R | A/R |
| 17_1 | 0.8 | 0.99 | 0.874 | R | A/R |
| 17_1 | 0.9 | 0.99 | 0.874 | A | A |
| 17_1 | 0.85 | 0.96 | 0.874 | A | A/R |
| 17_1 | 0.9 | 0.96 | 0.874 | A | A |
| 16_3 | 0.8 | 0.9 | 0.879 | A | A/R |
| 16_3 | 0.8 | 0.94 | 0.879 | A | A/R |
| 16_3 | 0.85 | 0.99 | 0.879 | A | A/R |

| 16_3 | 0.9 | 0.99 | 0.879 | A | A |
|------|------|------|-------|---|-----|
| 16_3 | 0.75 | 0.9 | 0.879 | R | A/R |
| 16_3 | 0.85 | 0.96 | 0.879 | A | A/R |
| 16_3 | 0.8 | 0.96 | 0.879 | A | A/R |
| 16_3 | 0.8 | 0.99 | 0.879 | A | A/R |
| 16_3 | 0.75 | 0.99 | 0.879 | A | A/R |
| 16_3 | 0.75 | 0.96 | 0.879 | A | A/R |
| 16_3 | 0.9 | 0.96 | 0.879 | A | A |
| 16_3 | 0.85 | 0.9 | 0.879 | A | A/R |
| 16_3 | 0.9 | 0.94 | 0.879 | A | A |
| 16_3 | 0.85 | 0.94 | 0.879 | A | A/R |
| 16_3 | 0.75 | 0.94 | 0.879 | R | A/R |
| 15_2 | 0.85 | 0.9 | 0.905 | A | A/R |
| 15_2 | 0.9 | 0.94 | 0.905 | A | A/R |
| 15_2 | 0.85 | 0.94 | 0.905 | A | A/R |
| 15_2 | 0.9 | 0.96 | 0.905 | A | A/R |
| 15_2 | 0.8 | 0.96 | 0.905 | A | A/R |
| 15_2 | 0.85 | 0.96 | 0.905 | A | A/R |
| 15_2 | 0.85 | 0.99 | 0.905 | A | A/R |
| 15_2 | 0.9 | 0.99 | 0.905 | A | A/R |
| 15_2 | 0.8 | 0.94 | 0.905 | A | A/R |
| 15_2 | 0.75 | 0.94 | 0.905 | R | A/R |
| 15_2 | 0.75 | 0.9 | 0.905 | R | R |
| 15_2 | 0.8 | 0.9 | 0.905 | A | A/R |
| 15_2 | 0.75 | 0.99 | 0.905 | A | A/R |
| 15_2 | 0.8 | 0.99 | 0.905 | A | A/R |
| 15_2 | 0.75 | 0.96 | 0.905 | R | A/R |
| 18_4 | 0.75 | 0.99 | 0.907 | R | A/R |
| 18_4 | 0.8 | 0.94 | 0.907 | R | A/R |
| 18_4 | 0.85 | 0.99 | 0.907 | A | A/R |
| 18_4 | 0.8 | 0.96 | 0.907 | R | A/R |
| 18_4 | 0.85 | 0.94 | 0.907 | R | A/R |
| 18_4 | 0.8 | 0.9 | 0.907 | R | R |
| 18_4 | 0.85 | 0.9 | 0.907 | R | R |
| 18_4 | 0.75 | 0.94 | 0.907 | R | A/R |
| 18_4 | 0.9 | 0.99 | 0.907 | A | A/R |
| 18_4 | 0.75 | 0.96 | 0.907 | R | A/R |
| 18_4 | 0.8 | 0.99 | 0.907 | R | A/R |
| 18_4 | 0.75 | 0.9 | 0.907 | R | R |
| 18_4 | 0.9 | 0.96 | 0.907 | A | A/R |

| | | | | | |
|---|---|---|---|---|---|
| 18_4 | 0.85 | 0.96 | 0.907 | R | A/R |
| 18_4 | 0.9 | 0.94 | 0.907 | A | A/R |
| 17_3 | 0.9 | 0.94 | 0.914 | A | A/R |
| 17_3 | 0.8 | 0.99 | 0.914 | A | A/R |
| 17_3 | 0.75 | 0.99 | 0.914 | R | A/R |
| 17_3 | 0.8 | 0.96 | 0.914 | R | A/R |
| 17_3 | 0.85 | 0.96 | 0.914 | A | A/R |
| 17_3 | 0.8 | 0.94 | 0.914 | R | A/R |
| 17_3 | 0.85 | 0.9 | 0.914 | A | A/R |
| 17_3 | 0.9 | 0.99 | 0.914 | A | A/R |
| 17_3 | 0.85 | 0.94 | 0.914 | A | A/R |
| 17_3 | 0.75 | 0.94 | 0.914 | R | A/R |
| 17_3 | 0.8 | 0.9 | 0.914 | R | R |
| 17_3 | 0.75 | 0.96 | 0.914 | R | A/R |
| 17_3 | 0.9 | 0.96 | 0.914 | A | A/R |
| 17_3 | 0.85 | 0.99 | 0.914 | A | A/R |
| 17_3 | 0.75 | 0.9 | 0.914 | R | R |
| 16_1 | 0.85 | 0.9 | 0.918 | R | R |
| 16_1 | 0.8 | 0.99 | 0.918 | R | A/R |
| 16_1 | 0.8 | 0.9 | 0.918 | R | R |
| 16_1 | 0.9 | 0.94 | 0.918 | R | A/R |
| 16_1 | 0.85 | 0.94 | 0.918 | R | A/R |
| 16_1 | 0.8 | 0.94 | 0.918 | R | A/R |
| 16_1 | 0.85 | 0.99 | 0.918 | R | A/R |
| 16_1 | 0.9 | 0.99 | 0.918 | R | A/R |
| 16_1 | 0.75 | 0.9 | 0.918 | R | R |
| 16_1 | 0.85 | 0.96 | 0.918 | R | A/R |
| 16_1 | 0.9 | 0.96 | 0.918 | R | A/R |
| 16_1 | 0.75 | 0.94 | 0.918 | R | A/R |
| 16_1 | 0.8 | 0.96 | 0.918 | R | A/R |
| 16_1 | 0.75 | 0.99 | 0.918 | R | A/R |
| 16_1 | 0.75 | 0.96 | 0.918 | R | A/R |
| 18_2 | 0.75 | 0.99 | 0.918 | R | A/R |
| 18_2 | 0.9 | 0.96 | 0.918 | R | A/R |
| 18_2 | 0.8 | 0.99 | 0.918 | R | A/R |
| 18_2 | 0.9 | 0.94 | 0.918 | R | A/R |
| 18_2 | 0.85 | 0.9 | 0.918 | R | R |
| 18_2 | 0.85 | 0.94 | 0.918 | R | A/R |
| 18_2 | 0.9 | 0.99 | 0.918 | R | A/R |
| 18_2 | 0.8 | 0.96 | 0.918 | R | A/R |

| | | | | | |
|---|---|---|---|---|---|
| 18_2 | 0.85 | 0.96 | 0.918 | R | A/R |
| 18_2 | 0.75 | 0.94 | 0.918 | R | A/R |
| 18_2 | 0.8 | 0.9 | 0.918 | R | R |
| 18_2 | 0.8 | 0.94 | 0.918 | R | A/R |
| 18_2 | 0.85 | 0.99 | 0.918 | R | A/R |
| 18_2 | 0.75 | 0.96 | 0.918 | R | A/R |
| 18_2 | 0.75 | 0.9 | 0.918 | R | R |
| 15_1 | 0.85 | 0.96 | 0.927 | R | A/R |
| 15_1 | 0.85 | 0.9 | 0.927 | R | R |
| 15_1 | 0.9 | 0.99 | 0.927 | R | A/R |
| 15_1 | 0.75 | 0.94 | 0.927 | R | A/R |
| 15_1 | 0.8 | 0.9 | 0.927 | R | R |
| 15_1 | 0.9 | 0.96 | 0.927 | R | A/R |
| 15_1 | 0.8 | 0.94 | 0.927 | R | A/R |
| 15_1 | 0.75 | 0.96 | 0.927 | R | A/R |
| 15_1 | 0.8 | 0.99 | 0.927 | R | A/R |
| 15_1 | 0.9 | 0.94 | 0.927 | R | A/R |
| 15_1 | 0.75 | 0.9 | 0.927 | R | R |
| 15_1 | 0.85 | 0.99 | 0.927 | R | A/R |
| 15_1 | 0.8 | 0.96 | 0.927 | R | A/R |
| 15_1 | 0.75 | 0.99 | 0.927 | R | A/R |
| 15_1 | 0.85 | 0.94 | 0.927 | R | A/R |
| 18_3 | 0.75 | 0.96 | 0.930 | R | A/R |
| 18_3 | 0.8 | 0.99 | 0.930 | R | A/R |
| 18_3 | 0.85 | 0.9 | 0.930 | R | R |
| 18_3 | 0.9 | 0.96 | 0.930 | R | A/R |
| 18_3 | 0.9 | 0.94 | 0.930 | R | A/R |
| 18_3 | 0.85 | 0.94 | 0.930 | R | A/R |
| 18_3 | 0.75 | 0.94 | 0.930 | R | A/R |
| 18_3 | 0.8 | 0.9 | 0.930 | R | R |
| 18_3 | 0.85 | 0.99 | 0.930 | R | A/R |
| 18_3 | 0.9 | 0.99 | 0.930 | R | A/R |
| 18_3 | 0.8 | 0.96 | 0.930 | R | A/R |
| 18_3 | 0.75 | 0.9 | 0.930 | R | R |
| 18_3 | 0.85 | 0.96 | 0.930 | R | A/R |
| 18_3 | 0.8 | 0.94 | 0.930 | R | A/R |
| 18_3 | 0.75 | 0.99 | 0.930 | R | A/R |
| 15_4 | 0.9 | 0.94 | 0.941 | R | R |
| 15_4 | 0.85 | 0.94 | 0.941 | R | R |
| 15_4 | 0.8 | 0.96 | 0.941 | R | A/R |

| | | | | | |
|---|---|---|---|---|---|
| 15_4 | 0.8 | 0.9 | 0.941 | R | R |
| 15_4 | 0.85 | 0.9 | 0.941 | R | R |
| 15_4 | 0.8 | 0.99 | 0.941 | R | A/R |
| 15_4 | 0.75 | 0.94 | 0.941 | R | R |
| 15_4 | 0.75 | 0.9 | 0.941 | R | R |
| 15_4 | 0.9 | 0.96 | 0.941 | R | A/R |
| 15_4 | 0.85 | 0.96 | 0.941 | R | A/R |
| 15_4 | 0.75 | 0.99 | 0.941 | R | A/R |
| 15_4 | 0.85 | 0.99 | 0.941 | R | A/R |
| 15_4 | 0.75 | 0.96 | 0.941 | R | A/R |
| 15_4 | 0.9 | 0.99 | 0.941 | A | A/R |
| 15_4 | 0.8 | 0.94 | 0.941 | R | R |
| 17_4 | 0.75 | 0.99 | 0.941 | R | A/R |
| 17_4 | 0.75 | 0.96 | 0.941 | R | A/R |
| 17_4 | 0.9 | 0.96 | 0.941 | R | A/R |
| 17_4 | 0.8 | 0.99 | 0.941 | R | A/R |
| 17_4 | 0.85 | 0.96 | 0.941 | R | A/R |
| 17_4 | 0.8 | 0.94 | 0.941 | R | R |
| 17_4 | 0.85 | 0.99 | 0.941 | R | A/R |
| 17_4 | 0.75 | 0.94 | 0.941 | R | R |
| 17_4 | 0.9 | 0.94 | 0.941 | R | R |
| 17_4 | 0.85 | 0.94 | 0.941 | R | R |
| 17_4 | 0.75 | 0.9 | 0.941 | R | R |
| 17_4 | 0.8 | 0.96 | 0.941 | R | A/R |
| 17_4 | 0.8 | 0.9 | 0.941 | R | R |
| 17_4 | 0.85 | 0.9 | 0.941 | R | R |
| 17_4 | 0.9 | 0.99 | 0.941 | A | A/R |
| 16_0 | 0.85 | 0.9 | 0.954 | R | R |
| 16_0 | 0.8 | 0.99 | 0.954 | R | A/R |
| 16_0 | 0.9 | 0.94 | 0.954 | A | A/R |
| 16_0 | 0.9 | 0.99 | 0.954 | A | A/R |
| 16_0 | 0.9 | 0.96 | 0.954 | A | A/R |
| 16_0 | 0.85 | 0.96 | 0.954 | A | A/R |
| 16_0 | 0.8 | 0.96 | 0.954 | A | A/R |
| 16_0 | 0.85 | 0.94 | 0.954 | A | A/R |
| 16_0 | 0.8 | 0.94 | 0.954 | A | A/R |
| 16_0 | 0.85 | 0.99 | 0.954 | A | A/R |
| 16_0 | 0.75 | 0.96 | 0.954 | R | A/R |
| 16_0 | 0.75 | 0.9 | 0.954 | R | R |
| 16_0 | 0.75 | 0.94 | 0.954 | R | R |

| 16_0 | 0.8 | 0.9 | 0.954 | R | R |
|------|------|------|-------|---|-----|
| 16_0 | 0.75 | 0.99 | 0.954 | R | A/R |
| 17_0 | 0.85 | 0.94 | 0.968 | A | A/R |
| 17_0 | 0.85 | 0.96 | 0.968 | A | A/R |
| 17_0 | 0.8 | 0.94 | 0.968 | A | A/R |
| 17_0 | 0.85 | 0.99 | 0.968 | A | A/R |
| 17_0 | 0.75 | 0.9 | 0.968 | R | R |
| 17_0 | 0.75 | 0.94 | 0.968 | R | R |
| 17_0 | 0.8 | 0.96 | 0.968 | A | A/R |
| 17_0 | 0.8 | 0.9 | 0.968 | R | R |
| 17_0 | 0.75 | 0.99 | 0.968 | A | A/R |
| 17_0 | 0.75 | 0.96 | 0.968 | R | R |
| 17_0 | 0.85 | 0.9 | 0.968 | A | A/R |
| 17_0 | 0.8 | 0.99 | 0.968 | A | A/R |
| 17_0 | 0.9 | 0.96 | 0.968 | A | A/R |
| 17_0 | 0.9 | 0.94 | 0.968 | A | A/R |
| 17_0 | 0.9 | 0.99 | 0.968 | A | A/R |
| 15_0 | 0.8 | 0.9 | 0.984 | R | R |
| 15_0 | 0.9 | 0.96 | 0.984 | R | R |
| 15_0 | 0.85 | 0.96 | 0.984 | R | R |
| 15_0 | 0.9 | 0.99 | 0.984 | R | A/R |
| 15_0 | 0.8 | 0.94 | 0.984 | R | R |
| 15_0 | 0.8 | 0.99 | 0.984 | R | A/R |
| 15_0 | 0.75 | 0.9 | 0.984 | R | R |
| 15_0 | 0.75 | 0.99 | 0.984 | R | A/R |
| 15_0 | 0.85 | 0.99 | 0.984 | R | A/R |
| 15_0 | 0.9 | 0.94 | 0.984 | R | R |
| 15_0 | 0.85 | 0.94 | 0.984 | R | R |
| 15_0 | 0.8 | 0.96 | 0.984 | R | R |
| 15_0 | 0.85 | 0.9 | 0.984 | R | R |
| 15_0 | 0.75 | 0.96 | 0.984 | R | R |
| 15_0 | 0.75 | 0.94 | 0.984 | R | R |
| 16_2 | 0.9 | 0.99 | 0.987 | A | A/R |
| 16_2 | 0.85 | 0.96 | 0.987 | A | A/R |
| 16_2 | 0.75 | 0.94 | 0.987 | R | R |
| 16_2 | 0.8 | 0.9 | 0.987 | R | R |
| 16_2 | 0.8 | 0.94 | 0.987 | A | A/R |
| 16_2 | 0.85 | 0.99 | 0.987 | A | A/R |
| 16_2 | 0.8 | 0.96 | 0.987 | A | A/R |
| 16_2 | 0.75 | 0.96 | 0.987 | R | R |

| 16_2 | 0.75 | 0.9 | 0.987 | R | R |
|---|---|---|---|---|---|
| 16_2 | 0.8 | 0.99 | 0.987 | A | A/R |
| 16_2 | 0.9 | 0.96 | 0.987 | A | A/R |
| 16_2 | 0.75 | 0.99 | 0.987 | A | A/R |
| 16_2 | 0.9 | 0.94 | 0.987 | A | A/R |
| 16_2 | 0.85 | 0.94 | 0.987 | A | A/R |
| 16_2 | 0.85 | 0.9 | 0.987 | A | A/R |
| 18_1 | 0.9 | 0.96 | 0.993 | R | R |
| 18_1 | 0.8 | 0.94 | 0.993 | R | R |
| 18_1 | 0.85 | 0.99 | 0.993 | R | R |
| 18_1 | 0.9 | 0.99 | 0.993 | R | R |
| 18_1 | 0.8 | 0.99 | 0.993 | R | R |
| 18_1 | 0.75 | 0.9 | 0.993 | R | R |
| 18_1 | 0.85 | 0.96 | 0.993 | R | R |
| 18_1 | 0.75 | 0.99 | 0.993 | R | R |
| 18_1 | 0.75 | 0.96 | 0.993 | R | R |
| 18_1 | 0.9 | 0.94 | 0.993 | R | R |
| 18_1 | 0.85 | 0.94 | 0.993 | R | R |
| 18_1 | 0.85 | 0.9 | 0.993 | R | R |
| 18_1 | 0.75 | 0.94 | 0.993 | R | R |
| 18_1 | 0.8 | 0.9 | 0.993 | R | R |
| 18_1 | 0.8 | 0.96 | 0.993 | R | R |
| 18_0 | 0.75 | 0.94 | 0.994 | R | R |
| 18_0 | 0.8 | 0.9 | 0.994 | R | R |
| 18_0 | 0.9 | 0.99 | 0.994 | R | R |
| 18_0 | 0.9 | 0.96 | 0.994 | R | R |
| 18_0 | 0.85 | 0.96 | 0.994 | R | R |
| 18_0 | 0.8 | 0.94 | 0.994 | R | R |
| 18_0 | 0.85 | 0.99 | 0.994 | R | R |
| 18_0 | 0.75 | 0.96 | 0.994 | R | R |
| 18_0 | 0.8 | 0.99 | 0.994 | R | R |
| 18_0 | 0.75 | 0.9 | 0.994 | R | R |
| 18_0 | 0.9 | 0.94 | 0.994 | R | R |
| 18_0 | 0.75 | 0.99 | 0.994 | R | R |
| 18_0 | 0.8 | 0.96 | 0.994 | R | R |
| 18_0 | 0.85 | 0.94 | 0.994 | R | R |
| 18_0 | 0.85 | 0.9 | 0.994 | R | R |
| 17_2 | 0.75 | 0.99 | 0.998 | R | R |
| 17_2 | 0.75 | 0.96 | 0.998 | R | R |
| 17_2 | 0.9 | 0.94 | 0.998 | R | R |

| | | | | | |
|---|---|---|---|---|---|
| 17_2 | 0.85 | 0.9 | 0.998 | R | R |
| 17_2 | 0.85 | 0.94 | 0.998 | R | R |
| 17_2 | 0.85 | 0.96 | 0.998 | R | R |
| 17_2 | 0.8 | 0.94 | 0.998 | R | R |
| 17_2 | 0.75 | 0.94 | 0.998 | R | R |
| 17_2 | 0.8 | 0.96 | 0.998 | R | R |
| 17_2 | 0.8 | 0.9 | 0.998 | R | R |
| 17_2 | 0.85 | 0.99 | 0.998 | R | R |
| 17_2 | 0.75 | 0.9 | 0.998 | R | R |
| 17_2 | 0.9 | 0.96 | 0.998 | R | R |
| 17_2 | 0.9 | 0.99 | 0.998 | R | R |
| 17_2 | 0.8 | 0.99 | 0.998 | R | R |

## B.2.2 Real-world PCs

In the following table, the first column indicates the benchmark, the second indicates the time required for the test, and the third column indicates the test outcome. 'A' represents Accept and 'R' represents Reject.

Table 5: The Extended Table

| Benchmark | Teq(s) | Result |
|---|---|---|
| or-70-10-8-UC-10_0 | 23.2 | A |
| or-70-10-8-UC-10_1 | 22.72 | R |
| or-70-10-8-UC-10_2 | 22.92 | R |
| or-70-10-8-UC-10_3 | 22.87 | R |
| or-70-10-8-UC-10_4 | 22.78 | R |
| or-70-10-8-UC-10_5 | 23.06 | R |
| or-70-10-8-UC-10_6 | 22.99 | R |
| or-70-10-8-UC-10_7 | 22.93 | R |
| or-70-10-8-UC-10_8 | 22.82 | R |
| or-70-10-8-UC-10_9 | 22.82 | R |
| s641_15_7_0 | 33.66 | A |
| s641_15_7_1 | 33.4 | R |
| s641_15_7_2 | 33.45 | R |
| s641_15_7_3 | 33.32 | R |
| s641_15_7_4 | 33.51 | R |
| s641_15_7_5 | 33.21 | R |
| s641_15_7_6 | 33.46 | R |
| s641_15_7_7 | 33.23 | R |
| s641_15_7_8 | 33.61 | R |
| s641_15_7_9 | 33.51 | R |
| or-50-5-4_0 | 414.17 | A |
| or-50-5-4_1 | 414.84 | R |
| or-50-5-4_2 | 410.16 | R |
| or-50-5-4_3 | 414.15 | R |
| or-50-5-4_4 | 410.07 | R |
| or-50-5-4_5 | 412.27 | R |
| or-50-5-4_6 | 414.77 | R |
| or-50-5-4_7 | 415.19 | R |
| or-50-5-4_8 | 416.84 | R |
| or-50-5-4_9 | 408.59 | R |
| ProjectService3.sk_12_55_0 | 356.58 | A |

| | | |
|---|---|---|
| ProjectService3.sk_12_55_1 | 353.77 | R |
| ProjectService3.sk_12_55_2 | 355.93 | R |
| ProjectService3.sk_12_55_3 | 356.11 | R |
| ProjectService3.sk_12_55_4 | 356.15 | A |
| ProjectService3.sk_12_55_5 | 355.64 | R |
| ProjectService3.sk_12_55_6 | 357.89 | R |
| ProjectService3.sk_12_55_7 | 356.69 | R |
| ProjectService3.sk_12_55_8 | 353.36 | R |
| ProjectService3.sk_12_55_9 | 356.14 | R |
| s713_15_7_0 | 24.56 | R |
| s713_15_7_1 | 24.68 | R |
| s713_15_7_2 | 24.28 | R |
| s713_15_7_3 | 24.47 | R |
| s713_15_7_4 | 24.65 | R |
| s713_15_7_5 | 24.32 | R |
| s713_15_7_6 | 24.4 | R |
| s713_15_7_7 | 24.39 | R |
| s713_15_7_8 | 24.86 | A |
| s713_15_7_9 | 24.41 | R |
| or-100-10-2-UC-30_0 | 31.11 | R |
| or-100-10-2-UC-30_1 | 31.16 | R |
| or-100-10-2-UC-30_2 | 31.04 | R |
| or-100-10-2-UC-30_3 | 31.13 | R |
| or-100-10-2-UC-30_4 | 31.14 | R |
| or-100-10-2-UC-30_5 | 31.04 | A |
| or-100-10-2-UC-30_6 | 31.03 | R |
| or-100-10-2-UC-30_7 | 31.13 | R |
| or-100-10-2-UC-30_8 | 31.17 | R |
| or-100-10-2-UC-30_9 | 31.0 | R |
| s1423a_3_2_0 | 153.8 | R |
| s1423a_3_2_1 | 152.37 | R |
| s1423a_3_2_2 | 152.01 | R |
| s1423a_3_2_3 | 150.96 | R |
| s1423a_3_2_4 | 152.64 | R |
| s1423a_3_2_5 | 153.13 | A |
| s1423a_3_2_6 | 151.52 | R |
| s1423a_3_2_7 | 152.53 | R |
| s1423a_3_2_8 | 152.4 | R |
| s1423a_3_2_9 | 152.81 | R |
| s1423a_7_4_0 | 104.28 | R |
| s1423a_7_4_1 | 103.4 | R |
| s1423a_7_4_2 | 103.82 | R |
| s1423a_7_4_3 | 104.18 | R |
| s1423a_7_4_4 | 103.95 | R |
| s1423a_7_4_5 | 103.59 | R |
| s1423a_7_4_6 | 104.31 | R |
| s1423a_7_4_7 | 104.93 | R |
| s1423a_7_4_8 | 104.93 | A |
| s1423a_7_4_9 | 103.51 | R |
| or-50-5-10_0 | 282.09 | R |
| or-50-5-10_1 | 282.49 | R |
| or-50-5-10_2 | 279.63 | R |
| or-50-5-10_3 | 281.8 | R |
| or-50-5-10_4 | 280.69 | R |
| or-50-5-10_5 | 279.91 | R |
| or-50-5-10_6 | 283.05 | A |
| or-50-5-10_7 | 282.69 | R |
| or-50-5-10_8 | 279.65 | R |
| or-50-5-10_9 | 282.97 | R |

| | | |
|---|---|---|
| or-60-20-6-UC-20_0 | 359.89 | R |
| or-60-20-6-UC-20_1 | 362.3 | R |
| or-60-20-6-UC-20_2 | 363.1 | R |
| or-60-20-6-UC-20_3 | 363.11 | R |
| or-60-20-6-UC-20_4 | 362.76 | R |
| or-60-20-6-UC-20_5 | 358.76 | R |
| or-60-20-6-UC-20_6 | 363.32 | A |
| or-60-20-6-UC-20_7 | 358.41 | R |
| or-60-20-6-UC-20_8 | 358.8 | R |
| or-60-20-6-UC-20_9 | 362.8 | R |