# OpenReview forum: "Testing Probabilistic Circuits"
_NeurIPS.cc/2021/Conference — NeurIPS 2021 Poster_

### Official Review · Reviewer_KnQP · 2021-07-16

**Rating:** 7
**Confidence:** 3

**Summary:**

The contribution of this paper is a method for testing the closeness
of the probability distributions encoded by two different
probabalistic circuits (PCs). The closeness tester (called Teq),
assumes access to an approximate weighted counter and an approximate
weighted sampler. Teq is described (in words and pseudo-code) and the
bulk of the paper is devoted to proving (PAC-style) correctness of Teq
(ie Theorem 1) and characterising its time complexity (Theorem
2). Empirical results are given supporting the correctness of Teq and
the stated time complexity theoretical results on Teq.

**Limitations And Societal Impact:**

Yes, they point to applications in safety-critical testing and note that care must be taken when applying probabilistic results.

**Main Review:**

I think probabilistic circuits are an important probabilistic
representation. The authors tell us that "Determining the closeness of
models has applications in AI planning [12], bioinformatics [29, 31,
33] and probabilistic program verification [14, 21]." and I am
prepared to trust them, but it would have been better to include some
indication of how closeness results are used in applications (I
appreciate space is short!) This is briefly alluded to in the 'broader impact' section.

The goal of the paper is sufficiently important, so it remains to
consider whether the paper is technically correct and written clearly
enough. The Teq algorithm is clearly described, and the algorithm
itself is nice and straightforward: the hard work is ensuring that
Theorem 1 is correct (eg m, the number of samples, is big enough). I
did check the proof and found no errors (but max should be \max on
line 194); the empirical results provide further evidence that neither
the authors or I have missed errors in the proof.

typos:

138: lesser than -> less than
refs: fix capitalisation, eg bayesian -> Bayesian

**Time Spent Reviewing:**

2

---

> ### Author Response · Authors · 2021-08-07
> **Response to Reviewer KnQP**
>
> Thank you for carefully reviewing our paper!
>
> We will expand on the applications further in the Appendix.

---

> > ### Comment · Reviewer_KnQP · 2021-08-30
> > **response to author response**
> >
> > Yes, it's a good idea to have a bit more on applications.

---

### Official Review · Reviewer_3D6J · 2021-07-16

**Rating:** 6
**Confidence:** 3

**Summary:**

This paper proposed a polynomial-time algorithm to check the closeness of two probabilistic circuits (PCs) with respect to the total variation distance metric, given that the PCs support tractable approximate counting and sampling. Experiments on benchmarks show that the algorithm is efficient in practice.

**Limitations And Societal Impact:**

There shouldn't be any negative societal impact.

**Main Review:**

This paper identifies and studies a very interesting theoretical problem, namely testing the closeness of two PCs with respect to the total variation distance. To solve this task, the authors propose a polynomial-time algorithm, which could be potentially applied in other tasks such as structure-learning for PCs. The overall structure of the proofs and the intermediate results do not have obvious flaws, but the proofs are a little bit hard to follow in detail. I would recommend the authors to replace the actual proofs by the more intuitive sketch-proofs and move the details and computations to the appendix.

**Time Spent Reviewing:**

2

---

> ### Author Response · Authors · 2021-08-09
> **Thank you and Sketch of Proofs**
>
> Thank you for your careful review. We are glad that you appreciated the technical strengths of the work. We will provide a high-level sketch of the proofs in the final version to enhance their readability.

---

### Official Review · Reviewer_LJEd · 2021-07-20

**Rating:** 6
**Confidence:** 4

**Summary:**

This paper presents a statistical closeness test for evaluating whether the total variation distance between the probability distributions encoded by two circuits is below a lower threshold or above an upper threshold with a given confidence in polynomial time. The authors introduce the setting, present the algorithm and give a sample of the analysis by proving parts of the main result. The results are evaluated on public datasets.

**Limitations And Societal Impact:**

It would be good if the authors discuss how the structure of circuits might impact efficiency (apart from lowering the cost of counting/evaluation).

**Main Review:**

The problem addressed in this paper is certainly an interesting and relevant one. The paper motivates this well. It is clearly written, definitely not an easy read, but there is nothing particularly problematic about it. I could get through it within reasonable time and effort. I could follow the main flow of the arguments and at a high level and they seem sound.

Regarding the originality I think there are two points of view that can be expressed here.
From the point of view of testing structured distributions, the approach is quite novel given that the early stage of the state of the art on this problem. From the point of view of distribution closeness testing, the approach seems to follow relatively established lines, the contribution seems to be to account for the type of queries that are typically supported by common circuit families. One relatively important line of criticism is that the approach doesn't really account for the structured nature of distributions encoded by circuits. This is instead hidden under the oracle assumptions. Presumably, an algorithm that can unpack some of that structure would have greater significance on the field. This is not all bad though --- perhaps the fact can be interpreted as indicative that there is a lot more potential here, and perhaps this paper could provoke some follow-up research.

Finally, I think it's important to sweep through a gird of epsilon, eta, delta configurations evaluating the probability of failure (i.e. fixing delta, running the test multiple times and evaluating the actual probability of failure empirically) to see whether the bound provides a good qualitative characterization. It's also interesting to see how the runtime of the oracle implementation behaves as a function of the parameters.

A few nits from the paper:
* Definition 1 presents either a typo or a horrendous abuse of notation.  The assignment sigma is defined as a binary vector, yet the sum iterates over x in sigma as if sigma is a set. In the subsequent expression, the letter x is used to both represent the symbolic variable as a member of the set X (as in w(x)) and its value as a binary (as in x=1). A better notation here is to use \sum_x \in X { w(x) if \sigma(x) = 1 as in "the x'th coordinate of sigma is 1".
* the paper mentions that the problems in the dataset are publicly available. I did not find a reference to them if that is the case.

**Time Spent Reviewing:**

999

---

> ### Author Response · Authors · 2021-08-07
> **Response to Reviewer LJEd**
>
> We will seek to do the empirical evaluation suggested by the reviewer.
> We apologize for not providing reference to the source of benchmarks: https://zenodo.org/record/3793090#.YQw-iFMzZQI.
> As stated, we used WAPS (which uses D4) to compile instances into dDNNF. We will release the entire artifact upon publication.

---

> > ### Author Response · Authors · 2021-09-02
> > **Remaining Concerns?**
> >
> > Dear Reviewer,
> >
> >    Please let us know if there are any remaining concerns that we can address.

---

### Official Review · Reviewer_LYkj · 2021-07-21

**Rating:** 6
**Confidence:** 4

**Summary:**

This paper proposes a polytime probabilistic algorithm for testing the closetness of two probabilistic circuits (e.g., such as those based on weighted d-DNNFs).  This is the first such algorithm for a probabilistic circuit (whereas some have been known for their propositional analogues).


**Ethical Concerns:**

ok

**Limitations And Societal Impact:**

ok

**Main Review:**

The premise of the paper is simple: provide a polytime test whether two probabilistic circuits are equivalent.  To my knowledge, no such test existed before, and it should be useful.  For example, it could be useful to perform caching when learning a probabilistic circuit from data.  Hence, I believe there is agood chance this test could be the basis of useful future work.

The testing procedue itself seems relatively simple, and an analysis of its correctness is provided, tho I did not check the details.

I believe I have a question on the empirical evaluation, and how significant it is.  In particular, the paper says that in order to evaluate benchmarks with a known distance, a "one variable perturbation" scheme was used to synthesize a dataset.  It appears, according to the appendix, that this will perturb the weights of a single variable.  Is this a good enough test for the proposed algorithm?  Are we ultimately drawing samples from a probabilistic circuit, and then drawing enough of them (based on the marginal distribution of the individual variable) and then testing to see if that one particular pair of weights is close or not?

I would find an evaluation based on small enough d-DNNF circuits where the distance could be evaluated brute-force to be more compelling.  The instances could be simulated by learning two PCs from different datasets drawn from the same distribution.


**Time Spent Reviewing:**

2

---

> ### Author Response · Authors · 2021-08-07
> **Response to Reviewer LYkj**
>
> We want to highlight that the algorithm is accompanied by rigorous theoretical guarantees that make no assumptions about how the instances were generated.
>
> We chose the “one variable” perturbation scheme as it gave us precise total variation distance. Note that the Teq algorithm is not aware of the process of generating the benchmarks, and indeed its time complexity depends only on the parameters $\varepsilon, \eta, \delta$.
>
> We would be happy to add additional experiments with more variables being perturbed. We will also add an evaluation for small d-DNNF circuits in the final version. We thank the reviewer for their suggestion.

---

### Decision · Program_Chairs · 2021-09-27

**Decision:**

Accept (Poster)

**Comment:**

All reviewers recognized the value of having an approximate testing scheme to compute the total variation distance between two weighted d-DNNF formulas.

During the discussion, a number of suggestions to strengthen the work before being published has been highlighted. Namely, these include

  - fixing the presentation by properly comparing general probabilistic circuits (PCs, as defined in Choi et al.) and the weighted d-DNNF formulas used in this work, e.g., by discussing how the proposed methods can be extended from such weighted boolean circuits to general PCs
  - adding synthetic experiments with small-scale d-DNNF to compare against a ground truth
  - adding real-world experiments with PCs learned from data to show generality
  - performing a grid search over the several hyperparameters
  - refactoring the proofs as to provide a more high level introduction

As the authors promised some content/results but they did not provide it during the rebuttal phase, the paper is accepted subject to incorporating this content.